# Recovering Unbalanced Communities in the Stochastic Block Model with Application to Clustering with a Faulty Oracle*

**Chandra Sekhar Mukherjee** †
chandrasekhar.mukherjee@usc.edu

**Pan Peng** ‡
ppeng@ustc.edu.cn

**Jiapeng Zhang** †
jiapengz@usc.edu

## Abstract

The stochastic block model (SBM) is a fundamental model for studying graph clustering or community detection in networks. It has received great attention in the last decade and the balanced case, i.e., assuming all clusters have large size, has been well studied. However, our understanding of SBM with unbalanced communities (arguably, more relevant in practice) is still limited. In this paper, we provide a simple SVD-based algorithm for recovering the communities in the SBM with communities of varying sizes. We improve upon a result of Ailon, Chen and Xu [ICML 2013; JMLR 2015] by removing the assumption that there is a large interval such that the sizes of clusters do not fall in, and also remove the dependency of the size of the recoverable clusters on the number of underlying clusters. We further complement our theoretical improvements with experimental comparisons. Under the planted clique conjecture, the size of the clusters that can be recovered by our algorithm is nearly optimal (up to poly-logarithmic factors) when the probability parameters are constant.

As a byproduct, we obtain an efficient clustering algorithm with sublinear query complexity in a faulty oracle model, which is capable of detecting all clusters larger than $\tilde{\Omega}(\sqrt{n})$, even in the presence of $\Omega(n)$ small clusters in the graph. In contrast, previous efficient algorithms that use a sublinear number of queries are incapable of recovering any large clusters if there are more than $\tilde{\Omega}(n^{2/5})$ small clusters.

## 1 Introduction

Graph clustering (or community detection) is a fundamental problem in computer science and has wide applications in many domains, including biology, social science, and physics. Among others, the stochastic block model (SBM) is one of the most basic models for studying graph clustering, offering both a theoretical arena for rigorously analyzing the performance of different types of clustering algorithms, and synthetic benchmarks for evaluating these algorithms in practice. Since the 1980s (e.g., [19, 8, 15, 7]), there has been much progress towards the understanding of the statistical and computational tradeoffs for community detection in SBM with various parameter regimes. We refer to the recent survey [1] for a list of such results.

---

*Authors are in alphabetical order.

†Thomas Lord Department of Computer Science, University of Southern California. Research supported by NSF CAREER award 2141536.

‡School of Computer Science and Technology, University of Science and Technology of China. Research supported in part by NSFC grant 62272431 and "the Fundamental Research Funds for the Central Universities".

37th Conference on Neural Information Processing Systems (NeurIPS 2023).

In this paper, we focus on a very basic version of the stochastic block model.

**Definition 1.1** (The $\text{SBM}(n, k, p, q)$ model)**.** *In this model, given an $n$-vertex set $V$ with a hidden partition $V = \cup_{i=1}^{k} V_i$ such that $V_i \cap V_j = \emptyset$ for all $i \neq j$, we say a graph $G = (V, E)$ is sampled from $\text{SBM}(n, k, p, q)$, if for all pairs of vertices $v_i, v_j \in V$, (1) an edge $(v_i, v_j)$ is added independently with probability $p$, if $v_i, v_j \in V_\ell$ for some $\ell$; (2) an edge $(v_i, v_j)$ is added independently with probability $q$, otherwise.*

We are interested in the problem of *fully recovering* all or some of the clusters, given a graph $G$ that is sampled from $\text{SBM}(n, k, p, q)$. A cluster $V_i$ is said to be fully recovered if the algorithm outputs a set $S$ that is exactly $V_i$. Most of the previous algorithms on the full recovery of SBM either just work for the *nearly balanced* case (i.e., each cluster has size $\Omega(\frac{n}{k})$) when $k$ is small, say $k = o(\log n)$ (see e.g. [2]), or only work under the following assumption:

- *All* of the latent clusters are sufficiently large[4], i.e., for each $j$, $|V_j| = \tilde{\Omega}(\sqrt{n})$ (see e.g., [25, 6, 10, 9, 1, 28, 11]).

From a practical perspective, many real-world graphs may have many communities of different sizes, that is, large and small clusters co-exist in these graphs. This motivates us to investigate how to recover the communities in SBM if the latent communities have very different sizes. In particular, we are interested in *efficiently* recovering all the large clusters in the presence of small clusters. However, such a task can be quite difficult, as those small clusters may be confused with noisy edges. Indeed, most previous algorithms try to find all the $k$-clusters in one shot, which always computes some structures/information of the graph that are sensitive to noise (and small clusters). For example, the classical SVD-based algorithms (e.g., [25, 28]) first compute the first $k$ singular vectors of some matrix associated with the graph and then use these $k$ vectors to find clusters. Such singular vectors are sensitive to edge insertions or deletions (e.g. [13]). In general, this difficulty was termed by Ailon et al. [3] as "*small cluster barrier*" for graph clustering.

To overcome such a barrier, Ailon et al. [3, 4] proposed an algorithm that recovers all large latent clusters in the presence of small clusters under the following assumptions (see [4]),

- none of the cluster sizes falls in the interval $(\alpha/c, \alpha)$ for a number $\alpha \sim \Theta\left(\frac{\sqrt{p(1-q)n}}{p-q}\right)$ and $c > 1$ is some universal constant;

- there exists a large cluster, say of size at least $\Upsilon := \Theta\left(\max\left\{\frac{\sqrt{p(1-q)n}}{p-q}, \frac{k \log n}{(p-q)^2}\right\}\right)$.

The algorithm in [4] then has to exhaustively search for such a gap, and then apply a convex program-based algorithm to find a large cluster of size at least $\Upsilon$. As we discuss in the Appendix D. the assumption of the recoverable cluster being larger than $\Omega(\sqrt{p(1-q)n}/(p-q))$ is (relatively) natural as any polynomial time algorithm can only recover clusters of size $\Omega(\sqrt{n})$, under the planted clique conjecture. Still, two natural questions that remain are

1. *Can we break the small cluster barrier without making the first assumption on the existence of a gap between the sizes of some clusters?*

2. *Can we remove the dependency of the size of the recoverable cluster on the number $k$ of clusters? In particular, when $k \gg \sqrt{n}$, can we still recover a cluster of size $\tilde{\Omega}(\sqrt{n})$?*

The above questions are inherently related to the clustering problem under the faulty oracle model which was recently proposed by Mazumdar and Saha [23], as an instance from the faulty oracle model is exactly the graph that is sampled from SBM with corresponding parameters. Thus, it is natural to ask *if one can advance the state-of-the-art algorithm for recovering large clusters for the graph instance from the faulty oracle model using an improved algorithm for the SBM?*

## 1.1 Our contributions

We affirmatively answer all three questions mentioned above. Specifically, we demonstrate that clusters of size $\tilde{\Omega}(\sqrt{n})$ can be successfully recovered in both the standard SBM and the faulty oracle model, *regardless of* the number of clusters present in the graph. This guarantee surpasses any previous achievements in related studies. The practical implications of this finding are significant

---

[4]The assumption is sometimes implicit. E.g., in [28], in their Theorem 1, the lower bound on their parameter $\Delta$ implies a lower bound on the smallest cluster size.

since real-world networks often exhibit a substantial number of clusters (see e.g. [29]), varying in size from large to small.

### 1.1.1 Recovering large clusters in the SBM

We first provide a singular value decomposition (SVD) based algorithm, *without* assuming there is a gap between the sizes of some clusters, for recovering large latent clusters. Furthermore, the recoverability of the largest cluster is unaffected by the number of underlying clusters.

**Theorem 1.2** (Recovering one large cluster). *Let $G$ be a graph that is generated from the SBM($n, k, p, q$) with $\sigma = \max\left(\sqrt{p(1-p)}, \sqrt{q(1-q)}\right)$. If both of the following conditions are satisfied: (1) the size of the largest cluster, denoted by $s_{\max}$, is at least $s^* := \frac{2^{13} \cdot \sqrt{p(1-q) \cdot n} \cdot \log n}{(p-q)}$; (2) $\sigma^2 = \Omega(\log n / n)$. There exists a polynomial time algorithm that exactly recovers a cluster of size at least $\frac{s_{\max}}{7}$ with probability $1 - \frac{1}{n^2}$.*

We have the following remarks about Theorem 1.2. (1) By the assumption that $\sigma^2 = \Omega(\log n / n)$, we obtain that $p = \Omega(\frac{\log n}{n})$, which further implies that the expected degrees are at least logarithmic in $n$. This is necessary as exact recovery in SBM requires the node degrees to be at least logarithmic even in the balanced case (i.e. when all the clusters have the same size; see e.g. [1]). (2) In contrast to the work [4], our algorithm breaks the small cluster barrier and improves upon the result of [4] in the following sense: we do not need to assume there is a large interval such that the sizes of clusters do not fall in, nor do our bounds get affected with increasing number of small clusters. (3) As a byproduct of Theorem 1.2, we give an algorithm that improves a result of [28] on partially recovering clusters in the SBM in the balanced case. We refer to Appendix C for details.

In addition, the tradeoff of the parameters in our algorithm in Theorem 1.2 is nearly optimal up to polylogarithmic factors for constant $p$ and $q$ under the *planted clique conjecture* (see Appendix D).

**Recovering more clusters**. We can apply the above algorithm to recover even more clusters, using a "peeling strategy" (see [3]). That is, we first recover the largest cluster (under the preconditions of Theorem 1.2), say $V_1$. Then we can remove $V_1$ and all the edges incident to them and obtain the induced subgraph of $G$ on the vertices $V' := V \setminus \{V_1\}$, denoting it as $G'$. Note that $G'$ is a graph generated from SBM($n', k-1, p, q$) where $n' = n - |V_1|$. Then we can invoke the previous algorithm on $G'$ to find the largest cluster again. We can repeat the process until the we reach a point where the recovery conditions no longer hold on the residual graph. Formally, we introduce the following definition of *prominent* clusters.

**Definition 1.3** (Prominent clusters). *Let $V_1, \ldots, V_k$ be the $k$ latent clusters and $s_1, \ldots, s_k$ be the size of the clusters. WLOG we assume $s_1 \geq \cdots \geq s_k$. Let $k' \geq 0$ be the smallest integer such that one of the following is true. (1) $s_{k'+1} < \frac{2^{13} \cdot \sqrt{p(1-q)} \sqrt{\sum_{i=k'+1}^{k} s_i}}{(p-q)}$, (2) $\sigma^2 < \log(\sum_{i=k'+1}^{k} s_i) / (\sum_{i=k'+1}^{k} s_i)$. We call $V_1, \ldots, V_{k'}$ prominent clusters of $V$.*

By the above definition, Theorem 1.2, and the aforementioned algorithm, which we call RECURSIVECLUSTER, we can efficiently recover all these prominent clusters.

**Corollary 1.4** (Recovering all the prominent communities). *Let $G$ be a graph that is generated from the SBM($n, k, p, q$) model. Then there exists a polynomial time algorithm RECURSIVECLUSTER that correctly recovers all the prominent clusters of $G$, with probability $1 - o_n(1)$.*

**Experimental Comparisons.** We evaluate the performance of our algorithm in the simulation settings outlined in [4] and confirm its effectiveness. Moreover, the experiments conducted in [4] established that their gap constraint is an observable phenomenon. We demonstrate that our algorithm can accurately recover clusters even without this gap constraint. Specifically, we succeed in identifying large clusters in scenarios where there were $\Omega(n)$ single-vertex clusters, a situation where the guarantees provided by [4] are inadequate. We observed that simpler spectral algorithms, such as [28], also failed to perform well in this scenario. Furthermore, we observe that the run-time of our algorithm is significantly faster than the SDP based approach of [3, 4]. Finally, we present empirical evidence of the efficacy of our techniques beyond their theoretical underpinnings.

### 1.1.2 An algorithm for clustering with a faulty oracle

We apply the above algorithm to give an improved algorithm for a clustering problem in a faulty oracle model, which was proposed by [23]. The model is defined as follows:

**Definition 1.5.** *Given a set $V = [n] := \{1, \cdots, n\}$ of $n$ items which contains $k$ latent clusters $V_1, \cdots, V_k$ such that $\cup_i V_i = V$ and for any $1 \leq i < j \leq k$, $V_i \cap V_j = \emptyset$. The clusters $V_1, \ldots, V_k$ are unknown. We wish to recover them by making pairwise queries to an oracle $\mathcal{O}$, which answers if the queried two vertices belong to the same cluster or not. This oracle gives correct answer with probability $\frac{1}{2} + \frac{\delta}{2}$, where $\delta \in (0, 1)$ is a* bias *parameter. It is assumed that repeating the same question to the oracle $\mathcal{O}$, it always returns the same answer[5].*

Our goal is to recover the latent clusters *efficiently* (i.e., within polynomial time) with high probability by making as few queries to the oracle $\mathcal{O}$ as possible. One crucial limitation of all the previous polynomial-time algorithms ([23, 21, 27, 20, 14]) that make sublinear[6] number of queries is that they *cannot* recover large clusters, if there are at least $\tilde{\Omega}(n^{2/5})$ small clusters. Now we present our result for the problem of clustering with a faulty oracle.

**Theorem 1.6.** *In the faulty oracle model with parameters $n, k, \delta$, there exists a polynomial time algorithm* NOSIYCLUSTERING$(s)$*, such that for any $n \geq s \geq \frac{C \cdot \sqrt{n} \log^2 n}{\delta}$, it recovers all clusters of size larger than $s$ by making $\mathcal{O}(\frac{n^4 \log^2 n}{\delta^4 \cdot s^4} + \frac{n^2 \log^2 n}{s \cdot \delta^2})$ queries in the faulty oracle model.*

We remark that our algorithm works without the knowledge of $k$, i.e., the number of clusters. Note that Theorem 1.6 says even if there are $\Omega(n)$ small clusters, our efficient algorithm can still find all clusters of size larger than $\Omega(\frac{\sqrt{n} \log n}{\delta})$ with sublinear number of queries. We note that the size of clusters that our algorithm can recover is nearly optimal under the planted clique conjecture. Due to space constraints, all the missing algorithms, analyses, and proofs are deferred to Appendix E and F.

## 1.2 Our techniques

Now we describe our main idea for recovering the largest cluster in a graph $G = (V, E)$ that is generated from SBM$(n, k, p, q)$.

**Previous SBM algorithms** The starting point of our algorithm is a Singular Value Decomposition (SVD) based algorithm by [28], which in turn is built upon the seminal work of [25]. The main idea underlying this algorithm is as follows: Given the adjacency matrix $A$ of $G$, project the columns of $A$ to the space $A_k$, which is the subspace spanned by the first $k$ left singular vectors of $A_k$. Then it is shown that for appropriately chosen parameters, the corresponding geometric representation of the vertices satisfies a *separability* condition. That is, there exists a number $r > 0$ such that 1) vertices in the same cluster have a distance at most $r$ from each other; 2) vertices from different clusters have a distance at least $4r$ from each other. This is proven by showing that each projected point $P_{\mathbf{u}}$ is close to its center, which is point $\mathbf{u}$ corresponding to a column in the expected adjacency matrix $\mathrm{E}[A]$. There are exactly $k$ centers corresponding to the $k$ clusters. Then one can easily find the clusters according to the distances between the projected points.

The above SVD-based algorithm aims to find all the $k$ clusters at once. Since the distance between two projected points depends on the sizes of the clusters they belong to, the parameter $r$ is inherently related to the size $s$ of the smallest cluster. Slightly more formally, in order to achieve the above separability condition, the work [28] requires that the minimum distance (which is roughly $\sqrt{s}(p-q)$) between any two centers is at least $\Omega(\sqrt{n/s})$, which essentially leads to the requirement that the minimum cluster size is large, say $\Omega(\sqrt{n})$, in order to recover all the $k$ clusters.

**High-level idea of our algorithm** In comparison to the work [28], we do not attempt to find all the $k$ clusters at once. Instead, we focus on finding large clusters, one at a time. As in [28], we first project the vertices to points using the SVD. Then instead of directly finding the "perfect" clusters from the projected points, we first aim to find a set $S$ that is somewhat close to a latent cluster that is large enough. Formally, we introduce the following definition of $V_i$-*plural* set.

**Definition 1.7** (Plural set)**.** *We call a set $S \subset V$ as a $V_i$-plural set if (1) $|S \cap V_i| \geq 2^{13} \sqrt{n} \log n$; (2) For any $V_j \neq V_i$ we have $|S \cap V_j| \leq 0.1 \cdot |S \cap V_i|$.*

---

[5]This was known as *persistent noise* in the literature; see e.g. [17].

[6]Since there are $\Theta(n^2)$ number of possible queries, by "sublinear" number of queries, we mean the number of queries made by the algorithm is $o(n^2)$.

That is, a plural set contains sufficiently many vertices from one cluster and much fewer vertices from any other cluster.

Recall that $s^* := \frac{C\sqrt{p(1-q)\cdot n\cdot\log n}}{(p-q)}$ for $C = 2^{13}$, and $s_{\max} \geq s^*$. We will find a $V_i$-plural set for any cluster $V_i$ that is large enough, i.e., $|V_i| \geq \frac{s_{\max}}{7}$. To recover large clusters, our crucial observation is that it suffices to separate vertices of one large cluster from other *large* clusters, rather than trying to separate it from all the other clusters. This is done by setting an appropriate distance threshold $L$ to separate points from any two different and *large* clusters. Then by refining Vu's analysis, we can show that for any $u \in V_i$ with $|V_i| \geq \frac{s_{\max}}{7}$, the set $S$ that consists of all vertices whose projected points belong to the ball surrounding $u$ with radius $L$ is a $V_i$-plural set, for some appropriately chosen $L$. It is highly non-trivial to find such a radius $L$. To do so, we carefully analyze the geometric properties of the projected points. In particular, we show that the distances between a point and its projection can be bounded in terms of the $k'$-th largest eigenvalue of the expected adjacency matrix of the graph (see Lemma 2.2), for a carefully chosen parameter $k'$. To bound this eigenvalue, we make use of the fact that $A$ is a sum of many rank 1 matrices and Weyl's inequality (see Lemma 2.3). We refer to Section 2 for more details.

Now suppose that the $V_i$-plural set $S$ is independent of the edges in $V \times V$ (which is *not* true and we will show how to remedy this later). Then given $S$, we can run a statistical test to identify all the vertices in $V_i$. To do so, for any vertex $v \in V$, observe that the subgraph induced by $S \cup \{v\}$ is also sampled from a stochastic block model. For each vertex $v \in V_i$, the expected number of its neighbors in $S$ is

$$p \cdot |S \cap V_i| + q \cdot |S \setminus V_i| = q|S| + (p - q) \cdot |S \cap V_i|.$$

On the other hand, for each vertex $u \in V_j$ for some different cluster $V_j \neq V_i$, the expected number of its neighbors in $S$ is

$$p \cdot |S \cap V_j| + q \cdot |S \setminus V_j| = q|S| + (p - q) \cdot |S \cap V_j| \leq q|S| + (p - q) \cdot 0.1 \cdot |S \cap V_i|,$$

since $|S \cap V_j| \leq 0.1 \cdot |S \cap V_i|$ for any $V_j \neq V_i$. Hence there exists a $\Theta((p - q) \cdot |S \cap V_i|)$ gap between them. Thus, as long as $|S \cap V_i|$ is sufficiently large, with high probability, we can identify if a vertex belong to $V_i$ or not by counting the number of its neighbors in $S$.

To address the issue that the set $S$ does depend on the edge set on $V$, we use a two-phase approach: that is, we first randomly partition $V$ into two parts $U, W$ (of roughly equal size), and then find a $V_i$-plural set $S$ from $U$, then use the above statistical test to find all the vertices of $V_i$ in $W$ (i.e., $V \setminus U$), as described in IDENTIFYCLUSTER$(S, W, \overline{s})$ (i.e. Algorithm 4).

Note that the output, say $T_1$, of this test is also $V_i$-plural set. Then we can find all vertices of $V_i$ in $U$ by running the statistical test again using $T_1$ and $U$, i.e., invoking IDENTIFYCLUSTER$(T_1, U, \overline{s})$. Then the union of the outputs of these two tests gives us $V_i$. We note that there is correlation between $T_1$ and $U$, which makes our analysis a bit more involved. We solve it by taking a union bound over a set of carefully defined bad events; see the proof of Lemma 2.7.

### 1.3 Other related work

In [11] (which improves upon [12]), the author also gave a clustering algorithm for SBM that recovers a cluster at a time, while the algorithm only works under the assumption that all latent clusters are of size $\Omega(\sqrt{n})$, thus they do not break the "small cluster barrier".

The model for clustering with a faulty oracle captures some applications in *entity resolution* (also known as the *record linkage*) problem [16, 24], the signed edges prediction problem in a social network [22, 26] and the correlation clustering problem [5]. A sequence of papers has studied the problem of query-efficient (and computationally efficient) algorithms for this model [23, 21, 27, 20, 14]. We refer to references [23, 21, 27] for more discussions of the motivations for this model.

## 2 The algorithm in the SBM

We start by giving a high-level view of our algorithm (i.e., Algorithm 1). Let $G = (V, E)$ be a graph generated from SBM$(n, k, p, q)$. For a vertex $v$ and a set $T \subset V$, we let $N_T(v)$ denote the number of neighbors of $v$ in $T$.

We first preprocess (in Line 1) the graph $G$ by invoking Algorithm 2 PREPROCESSING, which randomly partitions $V$ into four subsets $Y_1, Y_2, Z, W$ such that each vertex is added to $Y_1, Y_2, Z, W$

with probability $1/8, 1/8, 1/4, 1/2$, respectively. Let $Y = Y_1 \cup Y_2, U = Y \cup Z$. See Figure 1 for a visual presentation of the partition. Let $\hat{A}$ (resp. $\hat{B}$) be the bi-adjacency matrix between $Y_1$ (resp. $Y_2$) and $Z$. This part is to reduce the correlation between some random variables in the analysis, similar to in [25] and [28]. Then we invoke (in Line 2) Algorithm 3 ESTMATINGSIZE to estimate the size of the largest cluster. It first samples $\sqrt{n}\log n$ vertices from $Y_2$ and then counts their number of neighbors in $W$. These counters allow us to obtain a good approximation $\bar{s}$ of $s_{\max}$.

We then repeat the following process to find a large cluster (or stop when the number of iterations is large enough). In Line 4–7, we sample a vertex $u \in Y_2$ and consider the column vector $\hat{\mathbf{u}}$ corresponding to $u$ in the bi-adjacency matrix $\hat{A}$ between $Y_2$ and $Z$. Then we consider the projection $P_{\hat{A}_{k'}}\hat{\mathbf{u}}$ of $\hat{\mathbf{u}}$ onto the subspace of the first $k'$ singular vectors of $\hat{A}$ for some appropriately chosen $k'$, and the set $S$ of all vertices $v$ in $Y_2$ whose projections are within distance $L/20$ from $\mathbf{p_u}$, for some parameter $L$. In Lines 9–15, we give a process that completely recovers a large cluster when $S$ is a plural set. More precisely, we first test if $|S| \geq \bar{s}/21$ and if so, we invoke Algorithm 4 to obtain $T_1 = \text{IDENTIFYCLUSTER}(S, W, \bar{s})$, which simply defines $T_1$ to be the set of all vertices $v \in W$ with $N_S(v) \geq q|S| + (p-q)\frac{\bar{s}}{56}$. Then we check (Line 10) if the set $T_1$ satisfies a few conditions to test if $u$ is indeed a good center (so that $S$ is a plural set) and test if $T_1 = V_1 \cap W$. If so, we then invoke $\text{IDENTIFYCLUSTER}(T_1, U, \bar{s})$ to find $V_1 \cap U$. Note that we use a two-step process to find $V_1$, as $N_S(u)$ is not a sum of independent events for $u \in U$.

---

**Algorithm 1** CLUSTER($G = (V, E), p, q$): Recovering one large cluster

---

1:   $\hat{A}, \hat{B}, Y_2, Y_1, Z, W \leftarrow \text{PREPROCESSING}(G, p, q)$
2:   $\bar{s} \leftarrow \text{ESTIMATINGSIZE}(G, p, q, W, Y_2)$
3: **for** $i = 1, \cdots, h = \sqrt{n}\log n$ **do**
4:      sample a vertex $u$ from $Y_2$
5:      $\mathbf{u} \leftarrow$ the column vector consisting of the edges between $u$ and $Z$
6:      $\mathbf{p_u} \leftarrow P_{\hat{A}_{k'}}\hat{\mathbf{u}}$, the projection of $\hat{\mathbf{u}}$ onto the subspace of the first $k'$ singular vectors of $\hat{A}$,
       where $k' = (p-q)\sqrt{n}/\sqrt{p(1-q)}$
7:      $S \leftarrow \{v \in Y_2: \|\mathbf{p_u} - \mathbf{p_v}\| \leq \frac{L}{20}\}$, where $\mathbf{p_v} \leftarrow P_{\hat{A}_{k'}}\hat{\mathbf{v}}$ and $L = \sqrt{0.004}(p-q)\sqrt{\bar{s}}$
8:      **if** $|S| \geq \frac{\bar{s}}{21}$ **then**
9:         Invoke $\text{IDENTIFYCLUSTER}(S, W, \bar{s})$ to get set $T_1$
10:        **if** $|T_1| \leq \frac{\bar{s}}{6}$ or $\exists v \in T_1$ s.t. $N_{T_1}(v) \leq (0.9p + 0.1q) \cdot |T_1|$ or $\exists v \in W \setminus T_1$ s.t.
        $N_{T_1}(v) \geq (0.9p + 0.1q) \cdot |T_1|$ **then**
11:           continue
12:        **else**
13:           Invoke $\text{IDENTIFYCLUSTER}(T_1, U, |T_1|)$ to obtain a set $T_2$
14:           Merge the two sets to form $T = T_1 \cup T_2$
15:           Return $T$.
16: Return $\emptyset$

---

## 2.1   The analysis

We first show that ESTIMATINGSIZE outputs an estimator $\bar{s}$ approximating the size of the largest cluster within a factor of 2 with high probability.

**Lemma 2.1.** *Let $\bar{s}$ be as defined in Line 6 of Algorithm 3. Then with probability $1 - n^{-8}$ we have $0.48 \cdot s_{\max} \leq \bar{s} \leq 0.52 \cdot s_{\max}$.*

Recall that $\hat{A}$ (resp. $\hat{B}$) is the bi-adjacency matrix between $Y_1$ (resp. $Y_2$) and $Z$. Let $A$ and $B$ be the corresponding matrices of expectations. That is, $\hat{A} = A + E$, where $E$ is a random matrix consisting of independent random variables with 0 means and standard deviations either $\sqrt{p(1-p)}$ or $\sqrt{q(1-q)}$.

For a vertex $u \in Y_1$, let $\hat{\mathbf{u}}$ and $\mathbf{u}$ represent the column vectors corresponding to $u$ in the matrices $\hat{A}$ and $A$ respectively (We define analogous notations for $\hat{B}$ and $B$ when $u \in Y_2$.). We let $e_u := \hat{\mathbf{u}} - \mathbf{u}$, i.e., $e_u$ is the random vector with zero mean in each of its entries. Recall that $\mathbf{p_u} = P_{\hat{A}_{k'}}\hat{\mathbf{u}}$.

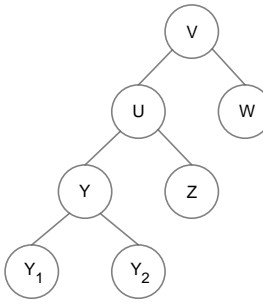

Figure 1: Partition of the vertices

---

**Algorithm 3** ESTIMATINGSIZE($G = (V, E), p, q, W, Y_2$): Estimating the size of the largest cluster

1: $s^* \leftarrow \frac{2^{13} \cdot \sqrt{p(1-q)} \cdot \sqrt{n} \cdot \log n}{(p-q)}$
2: **for** $i = 1, \cdots, h = \sqrt{n} \log n$ **do**
3:      sample $u_i$ from $Y_2$ uniformly at random.
4:      $N_W(u_i) \leftarrow$ # of neighbors of $u_i$ in $W$.
5: $u \leftarrow \arg\max N_W(u_i)$
6: $\bar{s} \leftarrow \frac{N_W(u) - q|W|}{(p-q)}$
7: **if** $\bar{s} \leq s^*/3$ **then**
8:      Exit(0)
9: **else**
10:      Return $\bar{s}$

---

**Algorithm 2** PREPROCESSING($G, p, q$): Partition and projection

1:    Randomly partitions $V$ into four subsets $Y_1, Y_2, Z, W$ such that each vertex is added to $Y_1, Y_2, Z, W$ with probability $1/8, 1/8, 1/4, 1/2$, respectively.
2: Let $Y = Y_1 \cup Y_2, U = Y \cup Z$.
3: Let $\hat{A}$ (resp. $\hat{B}$) be the bi-adjacency matrix between $Y_1$ (resp. $Y_2$) and $Z$.
4: Return $\hat{A}, \hat{B}, Y_2, Y_1, Z, W$

---

**Algorithm 4** IDENTIFYCLUSTER($S, R, \bar{s}$): Finding a subcluster $R \cap V_i$ using a $V_i$-plural set $S$

1: $T \leftarrow \emptyset$
2: **for** each $v \in R$ **do**
3:      **if** $N_{v,S} \geq q|S| + (p-q)\frac{\bar{s}}{56}$ **then**
4:         add $v$ to $T$
5: Return $T$

---

Now we bound the distance between $P_{\hat{A}_{k'}} \hat{\mathbf{u}}$ and the expectation vector $\boldsymbol{u}$. We set $\varepsilon = 0.002$ in the following.

**Lemma 2.2.** *Follows the setting of Algorithm 2, we fix $Y_1, Y_2, Z, W$. For any vector $u \in Y_2$ and $k' \geq 1$ we have $\|P_{\hat{A}_{k'}}(\hat{\mathbf{u}}) - \boldsymbol{u}\| \leq \frac{1}{\sqrt{s_u}}\|(P_{\hat{A}_{k'}} - I)A\| + \|P_{\hat{A}_{k'}}(e_u).\|$*

*Furthermore, for some constant $C_2$, and $\varepsilon$ as described above we have*

   1. *$\|(P_{\hat{A}_{k'}} - I)A\| = \|(P_{\hat{A}_{k'}} - I)\hat{A} - (P_{\hat{A}_{k'}} - I)E\| \leq 2C_2\sigma\sqrt{n} + \lambda_{k'+1}(A)$ with probability $1 - \mathcal{O}(n^{-3})$ for a random $\hat{A}$, where $\lambda_t(A)$ is the $t$-th largest sigular value of $A$.*

   2. *For any set $V' \subset Y_2$ s.t. $|V'| \geq \frac{4 \log n}{\varepsilon^2}$, with probability $1 - n^{-8}$, we have $\|P_{\hat{A}_{k'}}(e_u)\| \leq \frac{1}{\varepsilon}\sigma\sqrt{k'}$ for at least $(1 - 2\varepsilon)$ fraction of the points $u \in V'$.*

We have the following result regarding the $t$-th largest singular value $\lambda_t(A)$ of $A$.

**Lemma 2.3.** *For any $t > 1$, $\lambda_t(A) \leq (p-q)n/t$.*

Now we introduce the a definition of good center, the ball of which induces a plural set.

**Definition 2.4** (Good center). *We call a vector $\hat{\mathbf{u}} \in \hat{B}$ a good center if it belongs to a cluster $V_i$ such that $|V_i| \geq \frac{s_{\max}}{4}$ and $\left\|P_{\hat{A}_{k'}}(e_u)\right\| \leq \frac{1}{\varepsilon}\sigma\sqrt{k'}$.*

That is, a good center is a vertex that belongs to a large cluster and has a low $\ell_2$ norm after the projection. Then by Lemma 2.2, we have the following corollary on the number of good centers.

**Corollary 2.5.** *If $s_{\max} \geq 16\sqrt{n} \log n$, then with probability $1 - n^{-8}$ there are $(1 - 2\varepsilon) \cdot s_{\max}$ many good centers in $V$.*

This implies that if we sample $\frac{100n \log n}{\bar{s}}$ many vertices independently at random, we shall sample a good center with probability $1 - n^{-8}$.

**Good center leads to plural set**   We show that if at line 4 a good center from a cluster $V_i$ is chosen, then the set $S$ formed in line 7 is a $V_i$-plural set. Recall that $L = \sqrt{0.004}(p-q)\sqrt{s}$. Let $L_\varepsilon := L$.

**Lemma 2.6.** *Let $u$ be a good center belonging to $V_i \cap Y_2$ and $S = \{v \in Y_2 : \|\mathbf{p_u} - \mathbf{p_v}\| \leq L_\varepsilon/20\}$. Then it holds with probability $1 - \mathcal{O}(n^{-3})$ that $|V_i \cap S| \geq \bar{s}/21$ and for any other cluster $V_\ell$ with $\ell \neq i$, $|S \cap V_\ell| \leq 1.05\varepsilon\bar{s}$. Thus $S$ is a $V_i$ plural set as $1/21 \cdot 1/10 \geq 1.05\varepsilon$.*

**Plural set leads to cluster recovery**  We now prove that given a plural set for a large cluster $V_i$, we can recover the whole cluster. This is done by two invocations of Algorithm 4.

**Lemma 2.7.** *Let $U, W$ be the random partition as specified in Algorithm 1. Let $S \subseteq Y_2$ be the $V_i$-plural set where $|V_i| \geq s_{\max}/4$. Let $T_1 := \text{IDENTIFYCLUSTER}(S, W, \bar{s})$ and $T := T_1 \cup \text{IDENTIFYCLUSTER}(T_1, U, \bar{s})$. Then with probability $1 - \mathcal{O}(n^{-3})$, it holds that $T_1 = V_i \cap W$, $T_1 \geq \frac{\bar{s}}{6}$ and $T = V_i$.*

**Testing if $T_1$ is a sub-cluster**  Since $S$ may not be a plural set, we show that we can test if $T_1 = W \cap V_i$ for some large cluster $V_i$ using the conditions of Line 10 of Algorithm 1.

**Lemma 2.8.** *Let $v$ be a good center from $V_i \cap Y_2$ such that $|V_i| \geq \frac{s_{\max}}{4}$ and let $S = \{u \in Y_2 : \|\mathbf{p_u} - \mathbf{p_v}\| \leq \frac{L_\varepsilon}{30}\}$. Let $T_1$ be the set returned by $\text{IDENTIFYCLUSTER}(S, W, \bar{s})$. Then with probability at least $1 - n^{-8}$, $|T_1| \geq \frac{\bar{s}}{6}$ and $N_{T_1}(u) \geq (0.9p + 0.1q)|T_1|$ for any $u \in T_1$ and $N_{T_1}(u) \leq (0.9p + 0.1q)|T_1|$ for any $u \in W \setminus T_1$.*

Finally, we show that if the set $T_1 \neq V_i \cap W$ for some large cluster $V_i$, then it satisfies one of the conditions at line 10 of Algorithm 1. Together with the previous results this guarantees correct recovery of a large set at every round.

**Corollary 2.9.** *Let $T_1 = \text{IDENTIFYCLUSTER}(S, W, \bar{s})$ be a set such that $T_1 \neq V_i \cap W$ for any underlying community $V_i$ of size $|V_i| \geq s_{\max}/7$. Then with probability $1 - n^{-8}$ either $|T_1| \leq \frac{\bar{s}}{6}$ or there is a vertex $u \in T_1$ such that $N_{T_1}(u) \leq (0.9p + 0.1q)|T_1|$.*

**Remark 2.10.** *Note that in Lemma 2.8 and Corollary 2.9, the quantity $N_{T_1}(u)$ for any $u \in T_1$ is a sum of independent events. This is because the event that a vertex in $v \in W$ is chosen in $T_1$ is solely based on $N_u(S)$, where $S \cap T = \emptyset$. Thus, for any $u_1, u_2 \in T$, there is an edge between them (as per underlying cluster identities) independent of other edges in the graph.*

The proofs of the above results are deferred to Appendix 2.1.

Now we are ready to prove Theorem 1.2.

**Proof of Theorem 1.2**  By the precondition, we have that $s_{\max} \geq s^*$. First, in Line 2, Lemma 2.1 guarantees that $0.48 s_{\max} \leq \bar{s} \leq 0.52 s_{\max}$. By Corollary 2.5 and the fact that we iteratively sampled vertices $\Omega(\sqrt{n} \log n)$ times, with probability $1 - n^{-8}$, one such vertex $u$ is a good center. Given such a good center, by Lemma 2.6, we know with probability $1 - \mathcal{O}(n^{-3})$, a $V_i$-plural set is recovered on Line 7. Then by Lemma 2.7, given such a $V_i$-plural set, the two invocations of $\text{IDENTIFYCLUSTER}$ recovers the cluster $V_i$ with probability $1 - \mathcal{O}(n^{-3})$. Furthermore, Lemma 2.8 shows that if the sampled vertex $v$ is a good center, then with probability $1 - n^{-8}$ none of the conditions of line 10 are satisfied, and we are able to recover a cluster. On the other hand, Corollary 2.9 shows that if $T_1 \neq V_i \cap W$ for any large cluster $V_i$, ($V_i : |V_i| \geq s_{\max}/7$) then one of the conditions of line 10 is satisfied with probability $1 - n^{-8}$ and the algorithm goes to the next iteration to sample a new vertex in line 4. Taking a union bound on all the events for at most $\mathcal{O}(\sqrt{n} \log n)$ iterations guarantees that algorithm 1 finds a cluster of size $s_{\max}/7$ with probability $1 - \mathcal{O}(n^{-2})$. This completes the correctness of Algorithm 1.

## 3   The algorithm in the faulty oracle model

We describe the main ideas of our algorithm NOISYCLUSTERING for clustering with a faulty oracle. Let $V$ be the set of items that contains $k$ latent clusters $V_1, \ldots, V_k$ and $\mathcal{O}$ be the faulty oracle. Following the idea of [27], we first sample a subset $T \subseteq V$ of appropriate size and query $\mathcal{O}(u, v)$ for all pairs $u, v \in T$. Then apply our SBM clustering algorithm (i.e. Algorithm 1 CLUSTER) on the graph (with all the edges for the pairs that are reported to belong to the same cluster) induced by $T$ to obtain clusters $X_1, \ldots, X_t$ for some $t \leq k$. We can show that each of these sets is a subcluster of some large cluster $V_i$. Then we can use majority voting to find all other vertices that belong to $X_i$, for each $i \leq t$. That is, for each $X_i$ and $v \in V$, we check if the number of neighbors of $v$ in $X_i$ is at least $\frac{|X_i|}{2}$. In this way, we can identify all the large clusters $V_i$ corresponding to $X_i$, $1 \leq i \leq t$.

Furthermore, we can just choose a small subset of $X_i$ of size $O(\frac{\log n}{\delta^2})$ for majority voting to reduce query complexity. Then we can remove all the vertices in $V_i$'s and remove all the edges incident to them from both $V$ and $T$ and then we can use the remaining subsets $T$ and $V$ and corresponding subgraphs to find the next sets of large clusters. The algorithm NOISYCLUSTERING then recursively finds all the large clusters until we reach a point where the recovery condition on the current graph no longer holds. The pseudocode and the analysis of NOISYCLUSTERING are deferred to Appendix F.

## 4 Experiments

Now we exhibit various properties of our algorithms by running it on several unbalanced SBM instantiations and also compare our improvement w.r.t the state-of-the-art. We start by running our algorithm RECURSIVECLUSTER on the instances used by the authors of [4]. WLOG, we assume that $|V_1| \geq |V_2| \cdots \geq |V_k|$. We denote the algorithm in [4] by ACX.

| Exp. # | $n$ | $p, q$ | $k$ | Cluster sizes | Recovery by us | Recovery by ACX |
|--------|-----|--------|-----|---------------|----------------|-----------------|
| 1 | 1100 | 0.7, 0.3 | 4 | $\{800, 200, 80, 20\}$ | Largest cluster | All clusters |
| 2 | 3200 | 0.8, 0.2 | 5 | $\{800, 200, 200, 50, 50\}$ | Largest cluster | All clusters |
| 3 | 750 | 0.8, 0.2 | 4 | $\{500, 150, 70, 30\}$ | Largest cluster | *Incorrect* Recovery |
| 4 | 800 | 0.8, 0.2 | 4 | $\{500, 200, 70, 30\}$ | Two largest clusters | *Incorrect* Recovery |

Table 1: Comparing RECURSIVECLUSTER with ACX [4]

**Comparison with ACX** In Exp-1 (abbreviated for Experiment #1) and Exp-2, our algorithm recovers the largest cluster while ACX recovers all the clusters. This is because we have a large, *constant* lower bound on the size of the clusters we can recover. If we scale up the size of the clusters by a factor of 20 in those instances, then we are also able to recover all clusters.

**Overcoming the gap constraint in practice** Exp-3 is the "mid-size-cluster" experiment in [4]. In this case, ACX recovers the largest cluster completely, but only some fraction of the second-largest cluster, which is an incorrect outcome. In [4], the authors used this experiment to emphasize that their "gap-constraint" is not only a theoretical artifact but also observable in practice. In comparison, we recover the largest cluster while do not make any partial recovery of the rest of the clusters. In Exp-4, we modify the instance in Exp-3 by changing the size of the second cluster to 200. Note that this further reduces the gap, and ACX fails in this case as before. In comparison, we are able to recover both the largest and the second largest cluster. This exhibits that we are indeed able to overcome the experimental impact of the gap constraint observed in [4] in the settings of Table 1.

| Exp. # | $n$ | $p, q$ | $k$ | Cluster sizes | Recovery by us |
|--------|-----|--------|-----|---------------|----------------|
| 5 | 2900 | 0.7, 0.3 | 1000 | $\{1000, 903\} \cup \{1\}_{i=1}^{997}$ | Large clusters |
| 6 | 12300 | 0.85, 0.15 | 4 | $\{12000, 100, 100, 100\}$ | All clusters |

Table 2: Further Evaluation of RECURSIVECLUSTER

We then run some more experiments in the settings of Table 2 to describe other properties of our algorithms as well as demonstrate the practical usefulness of our "plural-set" technique.

**Many clusters** Exp-5 covers a situation where $k = \Omega(n)$ (specifically $n/3$), which can not be handled by ACX, as the size of the recoverable cluster in [4] is lower bounded by $k \log n/(p-q)^2 > n$. In comparison, our algorithm can recover the two main clusters. We also remark, in this setting, the spectral algorithm in [28] with $k = 1000$ can not geometrically separate the large clusters.

**Recovery of small clusters** Exp-6 describes a situation where the peeling strategy successfully recovers clusters that were smaller than $\sqrt{n}$ in the original graph. Once the largest cluster is removed, the smaller cluster then becomes recoverable in the residual graph. Finally, we discuss the usefulness of the plural set.

**Run-time comparison** Here, note that our method is a combination of a $(p-q)\sqrt{n}/\sqrt{p(1-q)}$ dimensional SVD projection, followed by some majority voting steps. Furthermore, we have

$(p-q)/\sqrt{p(1-q)} \le 2\sqrt{p}$. This implies that the time complexity of our algorithm is $\mathcal{O}\left(2\sqrt{p} \cdot n^{2.5}\right)$. In comparison, the central tool used in the algorithms by [3, 4] is an SDP relaxation, which scales as $\mathcal{O}(n^3)$. This implies that the asymptotic time complexity of our method is also an improvement on the state-of-the-art. We also confirm that the difference in the run-time becomes observable even for small values of $n$. For example, our algorithm recovers the largest cluster in Experiment 1 (with $n = 1100$) of table 1 in 1.4 seconds. In comparison, [3] recovers all 4 clusters, but takes 44 seconds.

**On the importance of plural sets** Recall that in Algorithm 1 (which is the core part of RECUR-SIVECLUSTER), we first obtain a plural-set $S$ in the partition $Y_2$ of $V$ (see Figure 1 to recall the partition). $S$ is not required to be $V_i \cap Y_2$ for any cluster $V_i$, but the majority of the vertices in $S$ must belong to a large cluster $V_i$ (which is the one we try to recover). We have the following observations:

1. In Exp-3 of Table 1, in the first round we recover a cluster $V_1$. Here in our first step, we recover a plural set $S$, where $S \subset V_1 \cap Y_2$. That is, we *do not recover* all the vertices of $V_1$ in $Y_2$ when forming the plural-set.

2. In Exp-4 of Table 1, in the second iteration we recover a cluster $V_2$. However, **the plural set $S \not\subset V_2$, and in fact contains a few vertices from $V_4$!** This is in fact the exact situation that motivates the plural-set method.

In both cases, the plural-set is then used to recover $S_1 := V_1 \cap W$ and $V_2 \cap W$ respectively, and then $S_1$ is used to recover the vertices of the corresponding cluster in $U$. Thus, our technique enables us to *completely* recover the largest cluster even though in the first round we may have some misclassifications. A more thorough empirical understanding of the Plural sets in different applications is an interesting future work.

We conclude our paper with some more discussion and future directions.

## 5 Conclusion

In this work, we design a spectral algorithm that recovers large clusters in the SBM model in the presence of arbitrary numbers of small clusters and compared to previous work, we do not require gap constraint in the size of consecutive clusters. Some interesting directions that remain open are as follows.

1. We note that both our algorithm and [4] require knowledge of the probability parameters $p$ and $q$ ([4] also need the knowledge of $k$, the number of clusters). Thus, whether parameter-free community recovery algorithms can be designed with similar recovery guarantees is a very interesting question.

2. Both our result (Algorithm 1) as well as [4] have a multiplicative $\log n$ term in our recovery guarantees. In comparison, the algorithm by Vu [28], which is the state-of-the-art algorithm in the dense case when "all" clusters are large, only has an additive logarithmic term. This raises the question of whether this multiplicative logarithmic factor can be further optimized when recovering large clusters in the presence of small clusters.

Additionally, we note that the constant in our recovery bound is quite large ($2^{13}$), and we have not made efforts to optimize this constant. We believe this constant can be optimized significantly, such as through a more careful calculation of the Chernoff bound in Theorem A.2.

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

# A  Preliminary Notations and Tools

**Notations for vectors.**  Let $\hat{M}$ be the adjacency matrix of the graph $G = (V, E)$ that is sampled from $\mathrm{SBM}(n, k, p, q)$. We denote by $M$ the matrix of expectations, where $M[i, j] = p$ if the $i$-th and $j$-th vertices belong to the same underlying cluster, and $M[i, j] = q$ otherwise. Going forward, we shall work with several sub-matrices of $\hat{M}$ and for any submatrix $M'$, we denote by $\widehat{M'}$ and $M'$ the random matrix and the corresponding matrix of expectations.

We also use the norm operator $\| \cdot \|$ frequently in this paper. We use the operator both for vectors and matrices. Given a vector $\mathbf{x} = (x_1, \dots, x_d)$, we let $\|\mathbf{x}\| := \sqrt{\sum_i x_i^2}$ denote its Euclidean norm. When the input is a matrix $M$, $\|M\|$ denotes the spectral norm of $M$, which is its largest singular value.

We describe the well-known Weyl's inequality.

**Theorem A.1** (Weyl's inequality). *Let $\hat{A} = A + E$ be a matrix. Then $\lambda_{t+1}(\hat{A}) \leq \lambda_{t+1}(A) + \|E\|$ where $\| \cdot \|$ is the spectral norm operator as described above.*

We will make use of the following general Chernoff Hoeffding bound.

**Theorem A.2** (Chernoff Hoeffding bound [18]). *Let $X_1, \dots, X_n$ be i.i.d random variables that can take values in $\{0, 1\}$, with $\mathrm{E}[X_i] = p$ for $1 \leq i \leq n$. Then we have*

1. $\mathbf{Pr}\left( \frac{1}{n} \sum_{i=1}^n X_i \geq p + \varepsilon \right) \leq e^{-D(p+\varepsilon \| p)n}$

2. $\mathbf{Pr}\left( \frac{1}{n} \sum_{i=1}^n X_i \leq p - \varepsilon \right) \leq e^{-D(p-\varepsilon \| p)n}$

Here $D(x\|y)$ is the KL divergence of $x$ and $y$. We recall the KL divergence between Bernoulli random variables $x, y$ $D(x\|y) = x \ln(x/y) + (1 - x) \ln((1 - x)/(1 - y))$. it is easy to see that If $x \geq y$, then $D(x\|y) \geq \frac{(x-y)^2}{2x}$, and $D(x\|y) \geq \frac{(x-y)^2}{2y}$ otherwise.

We also note down a random projection Lemma that we use in our proof.

**Lemma A.3** (Expected random projection [28]). *Let $P_{\hat{A}_{k'}}$ be a $k'$-dimensional projection matrix, and $e_u$ be an $n$ dimensional random vector where each entry is $0$ mean and has a variance of at most $\sigma^2$. Then we have $\mathrm{E}[\|P_{\hat{A}_{k'}}(e_u)\|^2] \leq \sigma^2 \cdot k'$.*

# B  Deferred Proofs from Section 2

We first give a general concentration bound concerning neighbors of vertices in the different partitions.

**Lemma B.1.** *Let $V$ be a set of $n$ vertices sampled according to the $SBM(n, k, p, q)$ model. Let $V' \subset V$ where the vertices in $V'$ are selected independently of each other. Let $V_i$ be a latent cluster with $V_i' = V_i \cap V'$. We denote by $N_{V'}(u)$ the number of neighbors of $u$ in $V'$. Then with probability $1 - \mathcal{O}(n^{-7})$ we have for every $u \in V_i'$,*

$$q|V'| + (p - q)|V' \cap V_i| - 16 \cdot \sqrt{p} \cdot \sqrt{n} \log n$$
$$\leq N_{V'}(u) \leq q|V'| + (p - q)|V' \cap V_i| + 48 \cdot \sqrt{p} \cdot \sqrt{n} \log n.$$

*Proof of Lemma B.1.*  We look at two different sums of random variables. The first is $N_{V_i'}(u)$ which is the sum of $|V_i \cap V'|$ many random $0 - 1$ variables with probability of $1$ being $p$. The second is $N_{V' \setminus V_i'}(u)$, which is the sum of $|V' \setminus V_i'|$ variables with probability of $1$ being $q$.

Then we have $\mathrm{E}[N_{V_i'}(u)] = p|V_i \cap V'|$ and $\mathrm{E}[N_{V' \setminus V_i'}(u)] = q|V' \setminus V_i'|$. Finally the Chernoff bound implies,

1. $\mathbf{Pr}\left( \frac{N_{V_i'}(u)}{|V_i'|} < p - \alpha \right) \leq e^{-D(p-\alpha \| p)|V_i'|}$. We fix $\alpha = \frac{8\sqrt{p}\sqrt{n}\log n}{|V_i'|}$ and then the term $D(p - \alpha \| p)|V_i'|$ evaluates to

$$D(p - \alpha \| p)|V_i'| \geq \frac{\alpha^2 |V_i'|}{2p} \geq \frac{8 \cdot p \cdot n \cdot 2 \log n \cdot |V_i'|}{|V_i'|^2 \cdot 2p} \geq \frac{8 \cdot \log n \cdot n}{|V_i'|} \geq 8 \log n.$$

This gives us

$$\mathbf{Pr}\left(N_{V_i'}(u) < p|V_i'| - 8\sqrt{p}\sqrt{n}\log n\right) \le n^{-8} \tag{1}$$

2. $\mathbf{Pr}\left(\frac{N_{V'\setminus V_i'}(u)}{|V'\setminus V_i'|} < q - \beta\right) \le e^{-D(q-\beta||q)|V'\setminus V_i'|}$. We fix $\beta = \frac{8\sqrt{p}\sqrt{n}\log n}{|V'\setminus V_i'|}$ and the term $D(q-\beta||q)|V'\setminus V_i'|$ evaluates to

$$D(q-\beta||q)|V'\setminus V_i'| \ge \frac{\beta^2}{2q} \ge \frac{8 \cdot p \cdot n \cdot 2\log n}{|V'\setminus V_i'|2q} \ge \frac{p \cdot 8\log n}{q} \cdot \frac{n}{|V'\setminus V_i'|} \ge 8\log n.$$

This gives us

$$\mathbf{Pr}\left(N_{V'\setminus V_i'}(u) < q|V'\setminus V_i'| - 8\sqrt{p}\sqrt{n}\log n\right) \le n^{-8} \tag{2}$$

Combining Equation (1) and (2) gives us

$$\mathbf{Pr}\left(N_{V_i'}(u) + N_{V'\setminus V_i'}(u) < p|V_i'| - 8\sqrt{p}\sqrt{n}\log n + q|V'\setminus V_i'| - 8\sqrt{p}\sqrt{n}\log n\right) \le 2n^{-8}$$
$$\implies \mathbf{Pr}\left(N_{V'}(u) < q|V'| + (p-q)|V_i'| - 16\sqrt{p}\sqrt{n}\log n\right) \le 2n^{-8}.$$

Now we study the event $N_{V'}(u) \ge q|V'| + (p-q)|V' \cap V_i| + 48 \cdot \sqrt{p} \cdot \sqrt{n}\log n$ again by breaking into two terms.

The probability bounds for two terms $N_{V_i'}(u)$ and $N_{V'\setminus V_i'}$ are $e^{-D(p+3\alpha||p)|V_i'|}$ and $e^{-D(q+3\beta||q)|V'\setminus V_i'|}$ respectively. Here note that we use $3\alpha$ instead of $\alpha$, to make calculations easier.

For the first case we have $D(p+3\alpha||p)|V_i'| \ge \frac{9\alpha^2|V_i'|}{2(p+3\alpha)}$. If $p \ge \alpha$ then $D(p+\alpha||p)|V_i'| \ge 9\alpha^2|V_i'|8p$ which implies we get the same bound as above. If $p < \alpha$ then $D(p+\alpha||p)|V_i'| \ge \frac{9\alpha^2|V_i'|}{8\alpha} \ge \alpha|V_i'|$. Now we have $\alpha = \frac{8\sqrt{p}\sqrt{n}\log n}{|V_i'|}$. Since $p = \Omega(\log n/n)$ we have $\alpha|V_i'| \ge 8\log n$. Combining we get that $e^{-D(p+2\alpha||p)|V_i'|} \le n^{-8}$.

Next we analyze $D(q+3\beta||q)|V'\setminus V_i'| \ge \frac{9\beta^2|V'\setminus V_i'|}{2(q+3\beta)}$. As before, if $q \ge \beta$ we have $D(q+3\beta||q)|V'\setminus V_i'| \ge \frac{9\beta^2|V_i|}{8q}$ and the result follows as before. Otherwise $D(q+3\beta||q)|V'\setminus V_i'| \ge \frac{9\beta^2|V'\setminus V_i'|}{8\beta} \ge \beta|V'\setminus V_i'| \ge 8\sqrt{p}\sqrt{n}\log n \ge 8\log n$, which completes the proof.

$\square$

Then Lemma 2.1 can be proved as follows.

*Proof of Lemma 2.1.* We know that $s_{\max} \ge \frac{2^{13} \cdot \sqrt{p(1-q)}\sqrt{n}\log n}{p-q}$ from the problem definition. Let the corresponding cluster be $V_i$. Then a simple application of Hoeffding bounds gives us that with probability $1 - n^{-8}$, $0.51 \cdot s_{\max} \ge |V_i \cap W| \ge 0.49 \cdot s_{\max}$.

Furthermore, we are interested in the regime where $p \le 3/4$ so $\sqrt{1-q} \ge 1/2$.

Then if we sample $u \in |V_i \cap Y_2|$, Lemma B.1 states that with probability $1 - n^{-8}$,

$$|N_W(u) - q|W| - (p-q)|V_i \cap W|| \le 48\sqrt{p}\sqrt{n}\log n \le \frac{(p-q) \cdot 96 \cdot \sqrt{p(1-q)}\sqrt{n}\log n}{(p-q)}$$
$$\implies |N_W(u) - q|W| - (p-q)|V_i \cap W|| \le \frac{(p-q)|V_i \cap W|}{100}$$

This implies with probability $1 - n^{-8}$, $q|W| + 1.01|V_i \cap W| \ge N_W(u) \ge q|W| + 0.99|V_i \cap W|$ which coupled with the fact $0.49 \le \frac{|V_i \cap W|}{|V_i|} \le 0.51$ implies that if we are able to sample a vertex from the largest cluster, we get an estimate of $s_{\max}$ as described.

Since $|V_i \cap Y_2| \geq 100\sqrt{n}\log n$, if we sample $\sqrt{n}\log n$ vertices, we sample a vertex $u$ from $V_i$ with probability $1 - n^{-8}$. Now, for vertices belonging to smaller clusters, the same bounds apply, and this implies that as long as we are able to sample a vertex from the largest cluster, we get an estimate of $s_{\max}$ between a factor of $0.48$ and $0.52$. $\qquad\square$

*Proof of Lemma 2.2.* These results follow directly from Vu [28] with some minor modifications. In their paper, Vu decomposes the matrix into $Y$ and $Z$. In comparison, we decompose the matrix to $U$ and $W$ first, and then $U$ is decomposed into $Y$ and $Z$. Thus the size of $Y$ and $Z$ in our framework is roughly half as compared to [28]. However, since the size of the clusters we are concerned about are all larger than $128 \cdot \sqrt{n}\log n$, the results follow in the same way with a change of a factor of 2.

Now we describe the results and how we deviate from Vu's analysis to get our result. For the first part, in [28, page 132] it was proved that for any fixed $\widehat{\mathbf{u}} \in \hat{B}$,

$$\|P_{\widehat{A}_k}(\widehat{\mathbf{u}}) - \boldsymbol{u}\| = \|P_{\widehat{A}_k}(\widehat{\mathbf{u}} - \boldsymbol{u}) + (P_{\widehat{A}_k} - I)\boldsymbol{u}\|$$

$$\leq \|P_{\widehat{A}_k}(e_u)\| + \|(P_{\widehat{A}_k} - I)\boldsymbol{u}\| \leq \|P_{\widehat{A}_k}(e_u)\| + \frac{1}{\sqrt{s_u}}\left\|(P_{\widehat{A}_k} - I)A\right\|.$$

Furthermore, it was proven (also in page 132 of [28]) that

$$\left\|(P_{\widehat{A}_k} - I)A\right\| = \left\|(P_{\widehat{A}_k} - I)\hat{A} - (P_{\widehat{A}_k} - I)E\right\|.$$

It was observed that $\left\|(P_{\widehat{A}_k} - I)\hat{A}\right\| \leq \lambda_{k+1}(\hat{A}) \leq \lambda_{k+1}(A) + \|E\| = \|E\|$ as $A$ has rank at most $k$; and $\left\|(P_{\widehat{A}_k} - I)E\right\| \leq \|E\|$. Then from Lemma 2.2 in [28] we have that with probability $1 - \mathcal{O}(n^{-3})$ $\|E\| \leq C_2\sigma n^{1/2}$ for some constant $C_2 > 0$.

Next, we observe that for any $k' \geq 1$, it still holds that

$$\|P_{\widehat{A}_{k'}}(\widehat{\mathbf{u}}) - \boldsymbol{u}\| = \|P_{\widehat{A}_{k'}}(\widehat{\mathbf{u}} - \boldsymbol{u}) + (P_{\widehat{A}_{k'}} - I)\boldsymbol{u}\|$$

$$\leq \|P_{\widehat{A}_{k'}}(e_u)\| + \|(P_{\widehat{A}_{k'}} - I)\boldsymbol{u}\| \leq \|P_{\widehat{A}_{k'}}(e_u)\| + \|(P_{\widehat{A}_{k'}} - I)A\|/\sqrt{s_u}.$$

Furthermore,

$$\begin{aligned}
\|(P_{\widehat{A}_{k'}} - I)A\| &= \|(P_{\widehat{A}_{k'}} - I)\hat{A} - (P_{\widehat{A}_{k'}} - I)E\| \\
&\leq \|(P_{\widehat{A}_{k'}} - I)\hat{A}\| + \|(P_{\widehat{A}_{k'}} - I)E\| \\
&\leq \lambda_{k'+1}(\hat{A}) + \|E\| \\
&\leq \lambda_{k'+1}(A) + 2\|E\|
\end{aligned}$$

Again, with probability at least $1 - 1/n^3$, $\|E\| \leq C_2\sigma\sqrt{n}$, which further implies that

$$\|(P_{\widehat{A}_{k'}} - I)A\| \leq 2C_2\sigma\sqrt{n} + \lambda_{k'+1}(A).$$

This is a simple but crucial step that removes our dependency on $k$, and allows us to treat all clusters of size $o(\sqrt{n})$ as noise.

Now, we analyze the first term. From Lemma A.3 we have $\mathrm{E}[\|P_{\widehat{A}_{k'}}(e_u)\|^2] \leq \sigma^2 k'$ for any $u \in Y_2$. Then for any $u$, Markov's inequality gives us $\mathbf{Pr}\left(\|P_{\widehat{A}_{k'}}(e_u)\| \geq \frac{\sigma\sqrt{k'}}{\varepsilon}\right) \leq \varepsilon$.

Now let us consider any set $V' \subset Y_2$ such that $|V'| \geq 16\sqrt{n}\log n$. For any $u \in V'$ we define $X_u$ to be the indicator random variable that gets 1 if $\|P_{\widehat{A}_{k'}}(e_u)\| \leq \frac{\sigma\sqrt{k'}}{\varepsilon}$, and 0 otherwise. Then $\mathrm{E}[X_u] \geq 1 - \varepsilon$. Now, since $V' \subset Y_2$, the variables $X_u$ are independent of each other (as $e_u$ are independent of each other). Then, using the fact that $|V'| \geq \frac{4\log n}{\varepsilon^2}$, the Chernoff bound gives us

$$\mathbf{Pr}\left(\sum_{u \in V'} X_u \leq (1 - \varepsilon)|V'| - \varepsilon|V'|\right) \leq e^{-\frac{2\varepsilon^2|V'|^2}{|V'|}} \leq n^{-8}$$

That is, with probability at least $1 - n^{-8}$, for at least $(1 - 2\varepsilon)$ fraction of the points $u \in V'$, $\|P_{\widehat{A}_{k'}}(e_u)\| \leq \frac{1}{\varepsilon}\sigma\sqrt{k'}$. $\qquad\square$

*Proof of Lemma 2.3.* Let there be $k$ many clusters $V_1, \ldots, V_k$ in the SBM problem. Then we define $a_i = |V_i \cap Z|$ and $b_i = |V_i \cap Y_1|$. Then we have that $A$ is an $n_1 \times n_2$ matrix where $n_1 = \sum_{i=1}^{k} a_i$ and $n_2 = \sum_{i=1}^{k} b_i$. The matrix $A$ can be then written as a sum of $k+1$ many rank 1 matrices:

$$A = \sum_{i=1}^{k} (p-q)M_i + qM_0$$

Here $M_0$ is the all 1 matrix, and $M_i$ is a block matrix with 1's in a $a_i \times b_i$ sized diagonal block. Since $M_i$ is a $a_i \times b_i$ block diagonal matrix of rank 1, with each entry being $(p-q)$, its singular value is $(p-q)\sqrt{a_i b_i}$. Now we define $A_1 = \sum_{i=1}^{k} (p-q)M_i$. As $M_i$'s are non-overlapping block diagonal matrices, the singular vectors of $M_i$ are also singular vectors of $A_1$, with the same singular values.

Thus, the sum of singular values of $A_1$ is $(p-q)\sum_{i=1}^{k} \sqrt{a_i b_i} \leq (p-q)\sqrt{n_1 n_2} \leq \frac{(p-q)n}{2}$, where the first inequality follows from the Cauchy-Schwarz inequality. Thus, for any $t \geq 1$

$$t \cdot \lambda_t(A_1) \leq \lambda_1(A_1) + \cdots \lambda_t(A_1) \leq \frac{(p-q)n}{2},$$

which gives $\lambda_t(A_1) \leq \frac{(p-q)n}{2t}$. Since $M_0$ has rank 1, $\lambda_2(q \cdot M_0) = 0$ and thus for $t > 1$ we have

$$\lambda_{t+1}(A) \leq \lambda_t(A_1) + \lambda_2(q \cdot M_0) \leq \frac{(p-q)n}{2t} \leq \frac{(p-q)n}{t+1},$$

where the first inequality follows from the Weyl's inequality. $\square$

The guarantee of obtaining a plural set is a consequence of Lemma B.2.

**Lemma B.2.** *Let $k' = \frac{(p-q)\sqrt{n}}{\sqrt{p(1-q)}}$ and $\varepsilon = 0.002$. Let $u \in Y_2$ be a good center belonging to $V_i$, then*

1. *There is a set $V_i' \subset Y_2 \cap V_i$ such that $|V_i'| \geq (1 - 2\varepsilon)|Y_2 \cap V_i|$ so that for all $v \in V_i'$, we have $\|P_{\widehat{A}_{k'}}(u - v)\| \leq \frac{L_\varepsilon}{30}$ with probability $1 - \mathcal{O}(n^{-3})$.*

2. *For any $V_j \neq V_i$ which is a $\varepsilon$-large cluster, there is a set $V_j' \subset V_j \cap Y_2$ s.t $|V_j'| \geq (1 - 2\varepsilon)|V_j \cap Y_2|$ so that for all $v \in V_j'$ we have $\|P_{\widehat{A}_{k'}}(u - v)\| \geq \frac{L_\varepsilon}{6}$ with probability $1 - \mathcal{O}(n^{-3})$.*

*Proof of Lemma B.2.* First note that $L_\varepsilon \geq \sqrt{2\varepsilon} \cdot 2^{11} \cdot \frac{(p-q)\cdot(p(1-q))^{1/4} \cdot n^{1/4} \cdot \log^{1/2} n}{(p-q)^{1/2}} \geq 132,000 \cdot (p-q)^{1/2} \cdot (p(1-q))^{1/4} \cdot n^{1/4} \cdot \sqrt{\log n}$.

When $u$ and $v$ belong to the same cluster we have $\|P_{\widehat{A}_{k'}}(\widehat{u} - \widehat{v})\| \leq \|P_{\widehat{A}_{k'}}(\widehat{u}) - u\| + \|P_{\widehat{A}_{k'}}(\widehat{v}) - v\|$.

Now, since $u$ is a good center, from Lemma 2.2 we have

$$\|P_{\widehat{A}_{k'}}(\widehat{u}) - u\| \leq \frac{1}{\varepsilon}\sigma\sqrt{k'} + \frac{1}{\sqrt{s_u}}\left(2C_2\sigma\sqrt{n} + \lambda_{k'+1}(\widehat{A})\right)$$

$$\leq \frac{1}{\varepsilon}\sqrt{p(1-q)}\sqrt{k'} + \frac{1}{\sqrt{s_u}}\left(2C_2\sqrt{p(1-q)}\sqrt{n} + \lambda_{k'+1}(A) + \|E\|\right)$$

$$\leq \frac{1}{\varepsilon}\sqrt{p(1-q)}\sqrt{k'} + \frac{1}{\sqrt{s_u}}\left(2C_2\sqrt{p(1-q)}\sqrt{n} + \lambda_{k'+1}(A) + C_2\sqrt{p(1-q)}\sqrt{n}\right)$$

$$\leq \frac{1}{\varepsilon}\sqrt{p(1-q)}\sqrt{k'} + \frac{1}{\sqrt{s_u}}\left(3C_2\sqrt{p(1-q)}\sqrt{n} + \frac{(p-q)n}{k'}\right) \quad \text{[Substituting } \lambda_{k'+1}(A) \text{ from Lemma 2.3]}$$

$$\leq \frac{1}{\varepsilon}(p(1-q))^{1/4}(p-q)^{1/2}n^{1/4} + 4C_2(\varepsilon)^{-1/2}(p-q)^{1/2}(p(1-q))^{1/4}n^{1/4}\log^{-1/2} n$$

$$\leq \frac{2}{\varepsilon}(p(1-q))^{1/4}(p-q)^{1/2}n^{1/4}$$

$$\leq \frac{10,000 L_\varepsilon}{132,000\log^{1/2} n} \leq \frac{L_\varepsilon}{60}, \quad \text{for } n \geq 64$$

Now from Lemma 2.2 we also know that at least $(1-2\varepsilon)$ fraction of the vertices $v \in V_i \cap Y_2$ are also "good centers" with probability $1-n^{-8}$. For all such vertices $\|P_{\widehat{A}_{k'}}(\widehat{\mathbf{u}}-\widehat{\mathbf{v}})\| \leq 2\|P_{\widehat{A}_{k'}}(\widehat{\mathbf{u}}-\boldsymbol{u})\| \leq \frac{L_\varepsilon}{30}$ with probability $1-n^{-3}$.

On the other hand when they belong to different clusters we have

$$\|P_{\widehat{A}_{k'}}(\widehat{\mathbf{u}}-\widehat{\mathbf{v}})\| \geq \|\boldsymbol{u}-\boldsymbol{v}\| - \|P_{\widehat{A}_{k'}}(\widehat{\mathbf{u}}-\boldsymbol{u})\| - \|P_{\widehat{A}_{k'}}(\widehat{\mathbf{v}}-\boldsymbol{v})\|.$$

Since $V_j$ is a $\varepsilon$-large cluster, $|V_i| \geq 256\sqrt{n}\log n$ and thus $|V_j \cap Y_2| \geq 16\sqrt{n}\log n$ with probability $1-\mathcal{O}(n^{-8})$. In that case for at least $1-2\varepsilon$ fraction of points $\boldsymbol{v} \in V_j \cap Y_2$ we have $P_{\widehat{A}_{k'}}(\widehat{\mathbf{v}}-\boldsymbol{v}) \leq \frac{L_\varepsilon}{60}$.

Now $\|\boldsymbol{u}-\boldsymbol{v}\| \geq (p-q)\sqrt{s_u+s_v} \geq \frac{\sqrt{2\varepsilon}\cdot(p-q)\sqrt{s_{\max}}}{6} \geq \frac{\sqrt{2\varepsilon}\cdot(p-q)\sqrt{\bar{s}}}{\sqrt{0.52}\cdot 6} \geq \frac{L_\varepsilon}{5}$ with probability $1-n^{-8}$ from Lemma 2.1. Thus with probability $1-n^{-3}$ we get

$$\|P_{\widehat{A}_{k'}}(\boldsymbol{u}-\boldsymbol{v})\| \geq \|\boldsymbol{u}-\boldsymbol{v}\| - \|P_{\widehat{A}_{k'}}(\widehat{\mathbf{u}}-\widehat{\mathbf{v}})\| - \|P_{\widehat{A}_{k'}}(\widehat{\mathbf{v}}-\boldsymbol{v})\| \geq \frac{L_\varepsilon}{5} - \frac{L_\varepsilon}{60} - \frac{L_\varepsilon}{60} \geq \frac{L_\varepsilon}{6}$$

for $1-2\varepsilon$ fraction of points $v \in Y_2 \cap V_j$ for any $\varepsilon$-large cluster $V_j$. This completes the proof. $\qquad\square$

*Proof of Lemma 2.6.* By Lemma 2.1, we have $|V_i| \geq \frac{s_{\max}}{4} \geq \frac{\bar{s}}{2.1}$ with probability $1-n^{-8}$. Then the following events happen.

1. Since $u \in V_i \cap Y_2$ is a good center, for $1-2\varepsilon$ fraction of points $v$ in $V_i \cap Y_2$, $\|\mathbf{p_u}-\mathbf{p_v}\| \leq L_\varepsilon/30$ with probability $1-\mathcal{O}(n^{-3})$ as per Lemma B.2. All such points are selected to $S$. Furthermore, $|V_i \cap Y_2|$ is lower bounded by $|V_i|/9$ with probability $1-n^{-8}$. Therefore, with probability $1-\mathcal{O}(n^{-3})$, we have

$$|V_i \cap S| \geq (1-2\varepsilon)|V_i \cap Y_2| \geq (1-2\varepsilon)|V_i|/9 \geq (1-2\varepsilon)\bar{s}/20 \geq \bar{s}/21.$$

2. For other clusters $V_\ell$, if $|V_\ell| \geq \varepsilon \cdot s_{\max}$, we have $\|\mathbf{p_u}-\mathbf{p_v}\| \leq \frac{L_\varepsilon}{6}$ for only $2\varepsilon$ fraction of points $v$ in $V_\ell \cap Y_2$ from Lemma B.2. Thus $|S \cap V_\ell| \leq 2\varepsilon|V_\ell \cap Y_2|$. On the other hand $|V_\ell \cap Y_2| \leq |V_\ell|/6$. Thus, with probability $1-n^{-8}$ we have

$$|S \cap V_\ell| \leq 2\varepsilon|V_\ell|/6 \leq \frac{2\cdot\varepsilon\cdot s_{\max}}{6} \leq \frac{2\cdot\varepsilon\cdot\bar{s}}{0.48\cdot 6} \leq 0.7\varepsilon\cdot\bar{s}$$

3. Otherwise, if $V_\ell$ is such that $\varepsilon\cdot s_{\max} \geq |V_\ell| \geq \frac{\varepsilon}{2}\cdot s_{\max}$, then $|V_\ell \cap Y_2| \leq |V_\ell|/6$ with probability $1-n^{-8}$. Then $|S \cap V_\ell| \leq \varepsilon s_{\max}/6 \leq \varepsilon\bar{s}$.

4. Otherwise, if $|V_\ell| \leq \frac{\varepsilon}{2}\cdot s_{\max}$ then $|S \cap V_\ell| \leq |V_\ell| \leq \frac{\varepsilon}{2}\cdot s_{\max} \leq \frac{\varepsilon\bar{s}}{2\cdot 0.48} \leq \frac{\varepsilon\bar{s}}{0.96} \leq 1.05\cdot\varepsilon\cdot\bar{s}$.

Now, note that for any $V_\ell$ with $\ell \neq i$, it holds with probability $1-\mathcal{O}(n^{-3})$ that $|S \cap V_\ell| \leq 1.05\cdot\varepsilon\cdot\bar{s} \leq (21\cdot 1.05\cdot\varepsilon)\cdot\frac{\bar{s}}{21} \leq 0.05\cdot\frac{\bar{s}}{21}$.

$\qquad\square$

*Proof of Lemma 2.7.* We first show that if $S$ is a $V_i$-plural set with $V_i \geq s_{\max}/4$, then $T_1 = V_i \cap W$ where $T_1$ is the outcome of IDENTIFYCLUSTER$(S, W, \bar{s})$. Since $S$ is a $V_i$ plural set, for any vertex $v \in W \cap V_i$, from Lemma B.1 we have that with probability $1-\mathcal{O}(n^{-3})$,

$$N_S(v) \geq q|S| + (p-q)|V_i \cap S| - 48\sqrt{p}\sqrt{n}\log n$$

$$\implies N_S(v) \geq q|S| + (p-q)\cdot\frac{\bar{s}}{21} - \frac{48}{\sqrt{1-q}}\cdot\frac{(p-q)\cdot\sqrt{p(1-q)}\sqrt{n}\log n}{(p-q)}$$

$$\implies N_S(v) \geq q|S| + (p-q)\cdot\frac{\bar{s}}{21} - (p-q)\cdot\frac{96\cdot\sqrt{p(1-q)}\sqrt{n}\log n}{(p-q)}$$

$$\implies N_S(v) \geq q|S| + (p-q)\cdot\frac{\bar{s}}{21} - (p-q)\cdot\frac{96\cdot\bar{s}}{2^{13}}$$

$$\implies N_S(v) \geq q|S| + (p-q)\cdot\frac{\bar{s}}{28}$$

Now, let us consider the case when $v \in V_j \cap W$ where $j \neq i$. Then we know from Lemma 2.6 $|S \cap V_j| \leq 1.05\varepsilon\overline{s} \leq \frac{\overline{s}}{210}$. Then using Lemma B.1 we have that with probability $1 - \mathcal{O}(n^{-3})$

$$N_S(v) \leq q|S| + (p-q)|V_j \cap S| + 24\sqrt{p}\sqrt{n}\log n$$

$$\implies N_S(v) \leq q|S| + (p-q)\frac{\overline{s}}{210} + (p-q)\frac{24\overline{s}}{2^{13}}$$

$$\implies N_S(v) \leq q|S| + (p-q)\frac{\overline{s}}{128}$$

Now note that in the IDENTIFYCLUSTER$(S, W, \overline{s})$ algorithm, we select all vertices from $W$ that have $q|S| + (p-q) \cdot \frac{\overline{s}}{56}$ neighbors in $S$. Thus, the above analysis implies with probability $1 - \mathcal{O}(n^{-3})$ $T_1 = \text{IDENTIFYCLUSTER}(S, W, \overline{s}) = V_i \cap W$. Furthermore, since $|V_i| \geq s_{\max}/4$, we have $|V_i \cap W| \geq \frac{s_{\max}}{2.2} \geq \frac{\overline{s}0.48}{2.2} \geq \overline{s}/6$.

We then use $T_1$ as a plural set to recover $V_i \cap U$ so that we are able to recover all the vertices of $V_i$, but now $T_1$ and $U$ are not completely independent and thus we cannot proceed simply as before.

We overcome this by an union bound based argument. Let's consider $T'_i = V_i \cap W$ for any $i$ such that $T' \geq \overline{s}/6$. Then we have the following facts.

1. Let $u \in U \cap V_i$. Then $\text{E}[N_{T'_i}(u)] = p|T'|$. Then Lemma B.1 shows that $\mathbf{Pr}(N_T(u) \leq q|T'| + 0.99(p-q)|T'|) \leq n^{-10}$.

2. Similarly, let $u \in U \cap V_j$. Then $\mathbf{Pr}(N_{T'_i}(u) \geq q|T'| + 0.01(p-q)|T'|) \leq n^{-10}$.

If either of this is true for a vertex $u \in U$ then we call it a bad vertex w.r.t $T'_i$. Then a union bound over all $V_i$ and all $u \in U$ gives us that no vertex $u \in U$ is bad w.r.t any $T'_i$ with probability $1 - n^{-8}$.

Then we can make this argument for $T'_i = T_1$. Since $|V_i| > s_{\max}/4$, we have $|V_i \cap W| \geq V_i/3$ with probability $1 - n^{-8}$. Then with probability $1 - n^{-8}$ no vertex $u \in U$ is bad w.r.t $T_1$.

Then applying Lemma B.1 to $T_1$ w.r.t vertices in $U$ we get, with probability $1 - \mathcal{O}(n^{-3})$

1. If $v \in V_i \cap U$, then $N_{T_1}(v) \geq q|T_1| + (p-q)|T_1| - (p-q)\frac{T_1}{96}$.

2. If $v \in V_j \cap U$, then $N_{T_1}(v) \leq q|T_1| + (p-q)\frac{T_1}{96}$.

Thus IDENTIFYCLUSTER$(T_1, U, \overline{s})$ only selects the set of vertices $T_2$ in $V_i \cap U$. Then taking the union of $T_1$ and $T_2$ gives us $V_i$. $\qquad \square$

*Proof of Lemma 2.8.* Since $|V_i| > 256\sqrt{n}\log n$, and every vertex of $V_i$ will be assigned to $W$ with probability $1/2$, we have that $|V_i \cap W| \geq \frac{|V_i|}{2.5} \geq \frac{s_{\max}}{10} \geq \frac{\overline{s}}{0.52 \cdot 10} \geq \frac{\overline{s}}{6}$ with probability $1 - n^{-8}$. Furthermore if $|V_i| \geq \frac{s_{\max}}{4}$, then Lemma 2.7 shows $T_1 = V_i \cap W$ and $|T_1| \geq \frac{\overline{s}}{6}$.

Furthermore, for any vertex $u \in T_1 \cap \{v\}$, we can calculate $N_{T_1}(u)$ in the following way.

We have $\text{E}[N_{T_1}(u)] = p|T_1|$. Then a simple application of Lemma B.1 give us that with probability $1 - n^{-8}$, $N_{T_1}(u) \geq p|T_1| - (p-q)\frac{T_1}{96} \geq (0.9p + 0.1q)|T_1|$.

Similarly, since $T_1 = V_i \cap W$ for any vertex $u \in W \cap T_1$, we have $\text{E}[N_{T_1}(u)] = q|T_1|$ and Lemma B.1 implies that with probability $1 - n^{-8}$

$$N_{T_1}(u) \leq q|T_1| + (p-q)\frac{|T_1|}{96} \leq \frac{p|T_1|}{3} + \frac{2q|T_1|}{3} \leq (0.33p + 0.67q)|T_1| < (0.9p + 0.1q)|T_1|.$$

$\qquad \square$

*Proof of Corollary 2.9.* If $T_1$ is a pure subset of some $V_i$, such that $|V_i| \leq s_{\max}/7$, then with probability $1 - \mathcal{O}(n^{-8})$, $|Y_2 \cap V_i| \leq \overline{s}/6$. If $|T_1| < \frac{\overline{s}}{6}$, the first condition is satisfied.

Otherwise if $|T_1| \geq \frac{\bar{s}}{6}$ and $T_1$ is not a pure set, there exists $V_j$ such that $|T_1 \cap V_j| \leq \frac{|T_1|}{2}$. In that case for any vertex $v \in V_j \cap T_1$ we have $\mathrm{E}[N_{T_1}(v)] \leq q|T_1| + (p-q)\frac{T_1}{2}$ and Lemma B.1 implies that with probability $1 - n^{-8}$,

$$N_{T_1}(u) \leq q|T_1| + (p-q)\frac{|T_1|}{2} + (p-q)\frac{|T_1|}{96} \leq (0.5 + 1/96)p|T_1| + (0.5 - 1/96)q|T_1| < (0.9p + 0.1q)|T_1|.$$

Finally if $T_1 \subset V_i$ is a large pure set and $T_1 \neq V_i \cap W$, then for a vertex $v \in V_i \cap (W \setminus T_1)$ we have $N_{T_1}(v) \geq (0.9p + 0.1q)|T_1|$.

$\square$

## C  An improved algorithm in the balanced case

Our algorithm is built upon [28] and [25]. However, even in the balanced case, our algorithm improves a result of [28] on partially recovering clusters in the SBM. More precisely, we can use Theorem 1.2 to prove the following theorem.

**Theorem C.1.** *Let $G = (V, E)$ be sampled from $\mathrm{SBM}(n, k, p, q)$ for $\sigma^2 = \Omega(\log n/n)$ where size of each cluster is $\Omega(n/k)$. Then there exists a polynomial time algorithm that exactly recovers all clusters if $(p-q)\sqrt{\frac{n}{k}} > C'\sigma\sqrt{k}\log n$ for some constant $C'$.*

In [28] (see Lemma 1.4 therein), Vu gave an algorithm that partially recovers all the clusters in the sense that with probability at least $1 - \varepsilon$, each cluster output by the algorithm contains at $1 - \varepsilon$ fraction of any one underlying communities, for any constant $\varepsilon > 0$. For the balanced case, his result holds under the assumption that $\sigma^2 > C \log n/n$, and $(p-q)\sqrt{\frac{n}{k}} > C\sigma\sqrt{k}$. In comparison, we obtain a *full* recovery of all the clusters under Vu's partial recovery assumptions at the cost of an extra $\log n$ factor in the tradeoff of parameters.

*Proof of Theorem C.1.* We have $(p-q)\sqrt{n/k} > C'\sigma\sqrt{k}\log n$. Let $C' = 2C$. Since $p \leq 3/4$, we have $1 - p \geq 1/4$ and then $\sigma \geq \frac{\sqrt{p(1-q)}}{2}$. Thus $(p-q)\sqrt{n/k} > C\sqrt{p(1-q)}\sqrt{k}\log n$. This implies $k < \frac{(p-q)\sqrt{n}}{C\sqrt{p(1-q)}\log n}$ and $n/k > \frac{C \cdot \sqrt{p(1-q)}\sqrt{n}\cdot\log n}{p-q}$. That is the size of each cluster is at least $s^*$. Then we can run Algorithm 1 to recover one such cluster. Now, since the size of each cluster is same, we can run this iteratively $k$ times, recovering a cluster at each round with probability $1 - \mathcal{O}(n^{-2})$. Using union bound we get that we are able to recover all clusters with probability $1 - \mathcal{O}(kn^{-2}) = 1 - \mathcal{O}(n^{-1})$. $\square$

## D  Lower bounds

First, we show that our algorithm is optimal up to logarithmic factors when $p$ and $q$ are constant. To do so, we make use of the well-known planted clique conjecture.

**Conjecture D.1** (Planted clique hardness). *Given an Erdős-Rényi random graph $G(n, q)$ with $q = 1/2$, if we plant in $G(n, q)$ a clique of size $t$ where $t \in [3 \cdot \log n, o(\sqrt{n})]$, then there exists no polynomial time algorithm to recover the largest clique in this planted model.*

Under the planted clique conjecture, we note that there is no polynomial time algorithm for the SBM problem that recovers clusters of size $o(\sqrt{n})$ irrespective of the number $k$ of clusters present in the graph, for any constants $p$ and $q$. This can be seen by defining the partition of $V$ as $V = \cup_{i=1}^{k} V_i$, where $V_1$ is a clique of size $t = o(\sqrt{n})$, and $V_2, \cdots, V_k$ are singleton vertices, $k = n - t$. Finally, let $p = 1, q = \frac{1}{2}$. Then an algorithm for finding a cluster of size $o(\sqrt{n})$ in a graph $G$ that is sampled from the SBM with the above partition solves the planted clique problem.

Thus, the dependency of our algorithm in Theorem 1.2 on $n$ is optimal under the planted clique conjecture up to logarithmic factors.

The following result was given in [23], we give a proof here for the sake of completeness.

**Theorem D.2** ([23]). *Let $A$ be a polynomial time algorithm in the faulty oracle model with parameters $n, k, \delta$. Suppose that $A$ finds a cluster of size $t$ irrespective of the value of $k$. Then under the planted clique conjecture, it holds that $t = \Omega(\sqrt{n})$.*

*Proof.* Let $G$ be a graph generated from the planted clique problem with parameter $t$. Note that each potential edge in the size-$t$ clique, say $K$, appears with probability 1, and each of the remaining potential edges appear with probability $\frac{1}{2}$. Now we delete each edge in $G$ with probability $\frac{1}{3}$. Then the resulting graph can be viewed as an instance generated from the faulty oracle model with parameters $n, k = n - t + 1$ and $\delta = \frac{1}{3}$: there are $k$ clusters, one being $H$, and $n - t$ clusters being singleton vertices. Furthermore, each intra-cluster edge is removed with probability $\frac{1}{3}$ and each inter-cluster is added with probability $\frac{1}{2} \cdot (1 - \frac{1}{3}) = \frac{1}{3}$. If there is a polynomial time algorithm that recovers the cluster $H$, no matter how many queries it performs, then it also solves the planted clique problem with clique size $t$. Under the planted clique conjecture, $t = \Omega(\sqrt{n})$. $\qquad\square$

## E   High-level ideas of the algorithm for the faulty oracle

**Discussion about the previous algorithm in the faulty oracle model**   One crucial limitation of all the previous polynomial-time algorithms that make sublinear number of queries is that they *cannot* recover large clusters, if there are at least $\tilde{\Omega}(n^{2/5})$ small clusters. The reason is that the query complexities of all these algorithms are at least $\Omega(k^5)$, and if there are $\tilde{\Omega}(n^{2/5})$ small clusters, then $k = \tilde{\Omega}(n^{2/5})$, which further implies that these polynomial time algorithms have to make $\Omega(k^5) = \Omega(n^2)$ queries.

**Main ideas of our algorithm**   Now we apply our algorithm in the SBM to the faulty oracle model. Consider the faulty oracle model with and parameters $n, k, \delta$. Assume that the oracle $\mathcal{O}$ outputs '+' to indicate the queried two vertices belong to the same cluster, and '-' otherwise.

Observe that if we make queries on all pairs $u, v \in V$, then the graph $G$ that is obtained by adding all + edges answered by the oracle $\mathcal{O}$ is exactly the graph that is generated from the SBM($n, k, p, q$) with parameters $n, k, p = \frac{1}{2} + \frac{\delta}{2}$ and $q = \frac{1}{2} - \frac{\delta}{2}$. However, the goal is to recover the clusters by making *sublinear* number of queries, i.e., without seeing the whole graph.

We now describe our algorithm NOISYCLUSTERING (i.e., Algorithm 5) for clustering with a faulty oracle. Let $V$ be the items which contains $k$ latent clusters $V_1, \ldots, V_k$ and $\mathcal{O}$ be the faulty oracle. Following the idea of [27], we first sample a subset $T \subseteq V$ of appropriate size and query $\mathcal{O}(u, v)$ for all pairs $u, v \in T$. Then apply our SBM clustering algorithm (i.e. Algorithm 1 CLUSTER) on the graph induced by $T$ to obtain clusters $X_1, \ldots, X_t$ for some $t \leq k$. We can show that each of these sets is a subcluster of some large cluster $V_i$. Then we can use a majority voting to find all other vertices that belong to $X_i$, for each $i \leq t$. That is, for each $X_i$ and $v \in V$, we check if the number of neighbors of $v$ in $X_i$ is at least $\frac{|X_i|}{2}$. In this way, we can identify all the large clusters $V_i$ corresponding to $X_i$, $1 \leq i \leq t$. Furthermore, we note that we can choose a small subset of $X_i$ of size $O(\frac{\log n}{\delta^2})$ for majority voting to reduce query complexity. Then we can remove all the vertices in $V_i$'s and remove all the edges incident to them from both $V$ and $T$ and then we can use the remaining subsets $T$ and $V$ and corresponding subgraphs to find the next sets of large clusters. The algorithm NOISYCLUSTERING then recursively find all the large clusters until we reach a point where the recovery condition on the current graph no longer holds.

## F   The algorithm in faulty oracle model

Now we turn to the faulty oracle model and give the corresponding algorithm Algorithm 5.

To analyze the algorithm NOISYCLUSTERING (i.e., Algorithm 5), we first describe two results.

**Lemma F.1.** *Let $|V| = n$ and $V_i \subset V : |V_i| = s \geq \frac{C\sqrt{n} \cdot \log^2 n}{\delta}$ for some constant $C > 1$. If a set $T \subset V$ of size $\frac{16C^2 n^2 \log n}{\delta^2 s^2}$ is sampled randomly, then with probability $1 - n^{-8}$, we have $|T \cap V_i| \geq \frac{C\sqrt{|T|} \log |T|}{4\delta} \geq \frac{C \log n}{\delta^2}$.*

---

**Algorithm 5** NOISYCLUSTERING($V, \delta, s$): recover all clusters of size more than $s \geq s^*$

---

1: $V' \leftarrow V; t' \leftarrow 0$
2: Randomly sample a subset $T \subset V'$ of size $|T| = \frac{C^2 n^2 \log^2 n}{s^2 \delta^2}$
3: Query all pairs $u, v \in T$ and let $G[T]$ be graph on vertex set $T$ with only positive edges from the query answers
4: **for** each $\ell$ from 1 to $\lfloor n/s \rfloor$ **do**
5:     Apply CLUSTER($G[T], \frac{1}{2} + \delta, \frac{1}{2} - \delta$) to obtain a cluster $T_\ell$
6:     **if** $T_\ell = \emptyset$ **then**
7:         continue
8:     **else**
9:         $t' \leftarrow t' + 1$
10:         Find an arbitrary subset $T'_\ell \subseteq T_\ell$ of size $\frac{4 \log n}{\delta^2}$
11:         $C'_{t'} \leftarrow \{v \in V' \setminus T : N_{T'_\ell}(v) \geq |T'_\ell|/2\}$
12:         $C_{t'} \leftarrow T_\ell \cap C'_{t'}$
13:         $T \leftarrow T \setminus T_\ell$.
14:         $V' \leftarrow V \setminus C_{t'}$
15: Return $C_1, \cdots, C_{t'}$

---

*Proof.* We use Hoeffding bound to obtain these bounds. We have $|T| \geq \frac{16 C^2 n^2 \log^2 n}{\delta^2 s^2} \geq 16 \log^2 n$. For every vertex $u \in T$, we define $X_u$ as the indicator random variable which is 1 if $u \in V_i$.

Then $E[X_u] = |V_i|/|V|$. Thus applying Hoeffding bound we get

$$\mathbf{Pr}\left(\sum_{u \in T} X_u \leq \frac{0.5 \cdot |T||V_i|}{|V|}\right) \leq e^{-8 \log n} \leq n^{-8}$$

Now, substituting value of $|T|$ we get $\frac{0.5 \cdot |T||V_i|}{|V|} \geq \frac{8 \cdot C^2 \cdot n^2 \log^2 n \cdot s}{s^2 \cdot \delta^2 \cdot n} \geq \frac{4C \cdot n \cdot \log n}{s \cdot \delta} \cdot \frac{C \cdot \log n}{\delta} \geq \frac{C \cdot \sqrt{|T|} \cdot \log n}{\delta} \geq \frac{C\sqrt{|T|} \cdot \log |T|}{\delta}$. Furthermore, the last equation shows $\frac{0.5 \cdot |T||V_i|}{|V|} \geq \frac{C\sqrt{|T|} \cdot \log |T|}{\delta} \geq \frac{Cn \log n}{s \cdot \delta \cdot \delta} \geq \frac{C \log n}{\delta^2}$. Now the proof follows by noting that $|T \cap V_i| = \sum_{u \in T} X_u$. $\qquad \square$

**Lemma F.2.** *Let $V$ be partitioned into two sets $U$ and $W$, where each vertex $v \in V$ is independently assigned to either set with equal probability. Let $S \subset V_i \cap U$ be a set such that $|S| \geq \frac{4 \log n}{\delta^2}$. Then with probability $1 - \mathcal{O}(n^{-8})$, we have $N_S(u) \geq \frac{|S|}{2}$ for all $u \in V_i \cap W$, and $N_S(u) < \frac{|S|}{2}$ for all $u \in V_j \cap W$ for any $j \neq i$.*

*Proof.* Let $u \in V_i \cap W$. Then $E[N_S(u)] = (0.5 + \delta) \cdot |S|$. Then

$$\mathbf{Pr}(N_S(u) \leq (0.5 + \delta) \cdot |S| - \delta|S|) = \mathbf{Pr}(N_S(u) \leq 0.5|S|) \leq e^{-2\delta^2 |S|^2/|S|} \leq e^{-2\delta^2 |S|} \leq e^{-8 \log n}$$

The last inequality holds $|S| \geq 4 \log n/\delta^2$. Thus if $u \in V_i \cap W$ then $N_S(u) \geq 0.5|S|$ with probability $1 - n^{-8}$.

Similarly, if $u \notin V_i$, then with probability $1 - n^{-8}$ we have $N_S(u) \leq 0.5|S|$.

$\qquad \square$

### F.1 Proof of Theorem 1.6

Given $s$, first we randomly sample $n' = \frac{C^2 n^2 \log^2 n}{s^2 \delta^2}$ many vertices from $V$, and denote this set as $T$.

Then Lemma F.2 proves that for any cluster $V_i : |V_i| \geq s^*$, we have $|T_i| = |T \cap V_i| \geq \frac{C\sqrt{n'} \log n'}{\delta}$ with probability $1 - n^{-8}$. For any underlying cluster $V_i$, we denote $T_i = T \cap V_i$.

Next we query all the pair of edges for vertices in $T$, which amounts $\mathcal{O}\left(\frac{n^4 \log^2 n}{\delta^4 s^4}\right)$ queries. The resultant graph $G'$ is an SBM graph on $n'$ vertices with $p = 0.5 + \delta$ and $q = 0, .5 - \delta$.

Thus, if we run Algorithm 1 with parameters $G', 0.5 + \delta, 0.5 - \delta$, then Theorem 1.2 implies that we recover a cluster $T_i$ such that $|T_i| \geq \frac{Cn' \log n'}{\delta}$ with probability $1 - n^{-2}$.

Once we get such a set $T_i$, we can take $4 \log n / \delta^2$ many vertices from it, calling it a set $S$. Then for every vertex $v \in V \setminus T$, we obtain $N_S(v)$, which requires $|S|$ many queries, and select all vertices such that $N_S(u) \geq 0.5|S|$. Lemma F.1 shows that we recover $V_i \cap (V \setminus T)$ with probability $1 - n^{-8}$, together recovering $V_i$. Thus this step requires $4n \log n / \delta^2$ queries for each iteration.

Once we have recovered $V_i$, we can then remove $T_i$ from $T$ and run Algorithm 1 again on the residual graph, followed by the sample-and recovery step of Line 10. Note that once we remove a recovered cluster, all sets $T_j$ that satisfied the recovery requirement of Theorem 1.2 in the graph $G'$ defined on $T$, also satisfies it on the graph $G''$ defined on $T \setminus T_i$, and we do not need to sample any more edges.

Finally, there are at most $\delta^2 \sqrt{T}$ many clusters $T_i \in T$ such that $|T_i| \geq \sqrt{|T|} \log |T| / \delta^2$. Here we have $\delta^2 \sqrt{T} = \frac{Cn \log n}{s}$. This upper bounds the number of iterations and thus the number of times the voting system on Line 10 is applied.

Thus the query complexity is $\mathcal{O}\left(\frac{n^4 \log^2 n}{\delta^4 s^4} + \frac{n \log n}{s} \cdot \frac{4n \log n}{\delta^2}\right) = \mathcal{O}\left(\frac{n^4 \log^2 n}{s^4 \cdot \delta^4} + \frac{n^2 \log^2 n}{s \cdot \delta^2}\right)$. This finishes the proof of Theorem 1.6.

