# OpenReview forum: "Recovering Unbalanced Communities in the Stochastic Block Model with Application to Clustering with a Faulty Oracle"
_NeurIPS.cc/2023/Conference — NeurIPS 2023 poster_

### Official Review · Reviewer_5GRr · 2023-06-29

**Soundness:** 4 excellent
**Presentation:** 4 excellent
**Contribution:** 3 good
**Rating:** 7
**Confidence:** 3

**Summary:**

This paper studies a classic problem of recovering clusters in a random graph. Concretely, the authors consider the stochastic block model. Here there is an underlying graph on n nodes. The n nodes are partitioned into k unknown clusters. There is then an edge independently between any two nodes in the same cluster with probability p and between any two in different clusters with probability q < p.

This is an extensively studied problem and many algorithms have been designed that allow the recovery of all clusters of a reasonable size (somewhat larger than sqrt(n), which is anyway a requirement for computational efficiency under the planted clique conjecture). The previous state of the art allows recovering clusters under two assumptions (here simplified for clarity and brevity):

1. The clusters to recover have size at least max(sqrt(n), k)/(p-q).
2. There is a number alpha of about sqrt(n)/(p-q) such that no cluster has size in the interval [alpha/C, alpha] for a constant C.

The assumption that the cluster sizes are at least sqrt(n) for those to be recovered is natural as mentioned above. However, the dependency on k is unfortunate when there are many small clusters. These would prevent the recovery of medium sizes clusters when k >> sqrt(n). Secondly, the assumption about the empty interval is quite unnatural.

The main contribution of this work is to remove the dependency on k in 1. and to remove the assumption 2. all together.

The authors also present applications of their algorithm in the related problem of clustering with a faulty oracle. Here one can ask whether two nodes v, w are in the same cluster or not. One is then returned a noise answer. Here the paper also improves over the state of the art in terms of the cardinality of clusters that can be recovered.

**Strengths:**

-The problem studied is fundamental in graph clustering.
-Removing the dependency on k and the requirement of an empty interval of cluster sizes is significant and the algorithm guarantees of the algorithm much more natural than previously
-The authors have implemented their algorithm and compared experimentally to previous work. The comparison is overall in favour of the new algorithm.

**Weaknesses:**

-I know this is a theoretical contribution, and also the authors probably did not attempt to optimize constants that much, but a factor 2^13 in the guarantees is quite severe in practice. Hopefully and probably, this constant is smaller in practice.

**Questions:**

-Could you say a bit about the running time of your algorithm in practice compared to previous work?
-Can you comment on whether the 2^13 constant can be reduced to a more reasonable constant without too much effort?

**Limitations:**

Yes

---

> ### Author Rebuttal · Authors · 2023-08-09
>
> Thanks a lot for your thorough and detailed review! Please find the answers to the questions below.
>
>
>
> ### Run time comparison:
> From our experiments, we find our algorithm to be much faster compared to the state-of-the-art algorithms of Ailon, Chen, and Xu (JLMR 2015).
>
> For example, in experiment 3A of the JLMR 2015 paper, they mention that when dealing with an instance with $n=3500, k=4$, they need $182$ seconds to recover the clusters. In comparison, our method requires around $2.5$ seconds. Thus, this is another empirical improvement over the state-of-the-art algorithms!
>
>
> ### Value of constants:
> We acknowledge that $2^{13}$ might seem like a large constant, and we agree that it is an artifact of our current proof. However, based on extensive experimentation, we estimate that this value could be significantly reduced to around $10-20$, which we consider a considerable improvement. Moreover, we firmly believe that the constant $2^{13}$ can be further optimized in various aspects of our analysis through more meticulous calculations. Specifically, a tighter analysis of the ``low probability Chernoff bound'' that we employ could lead to more refined and improved results.

---

> > ### Comment · Reviewer_5GRr · 2023-08-14
> >
> > Thanks to the authors for their rebuttal. I would recommend the authors to mention these favourable runtime results in their final paper as well. I keep my positive score of the paper.

---

> > > ### Author Response · Authors · 2023-08-14
> > >
> > > Thank you again for the detailed review and the response! We will definitely add a paragraph discussing the runtime comparisons in the final version.

---

### Official Review · Reviewer_MiyQ · 2023-07-06

**Soundness:** 4 excellent
**Presentation:** 3 good
**Contribution:** 4 excellent
**Rating:** 7
**Confidence:** 3

**Summary:**

This work studies stochastic block models where blocks/clusters can have different sizes. It proposed a simple SVD algorithm which recovers communities in this setting. The main technical improvement of this work is that the assumption is removed which requires there to be a ‘size interval’ where no clusters appear.
A secondary result is a efficient clustering algorithm with sublinear query complexity.


**Strengths:**

-	This work is a clear improvement over the previous state-of-the-art. As I understand it, a key technical contribution of this work that might influence future work is instead of finding $k$ clusters as is done using the SVD approach, the algorithm first aims to find large clusters one-by-one. Although these are not perfect (they form a so-called plural set), using some non-trivial techniques perfect recovery can be obtained.
- Experiments on synthetic data indicate that the algorithm not only works well in theory but also in practice.
- The write-up of this work is excellent.

**Weaknesses:**

-	Given that the aim of the studied setting is to look at more realistic settings, I would have expected to find experimental results on real-world datasets as well. Although this work does provide better bounds for SBMs generated with differently sized clusters, SBMs still have a highly symmetric structure compared to real-world graphs. It would be interesting to see the performance of the proposed algorithm on some real-world graphs.

**Questions:**

-	How does the algorithm compare with respect to the previous work in terms of running time?
- In practice the Spectral Clustering algorithm performs well in practice on graphs with clusters of unbalanced size. Even though not many bounds are known of spectral clustering with respect to SBMs, did you try to compare your algorithm experimentally with Spectral Clustering?

---

> ### Author Rebuttal · Authors · 2023-08-09
>
> Thanks a lot for your thorough and detailed review! Please find the answers to your questions below.
>
> ### Run time comparison:
> Regarding the asymptotic running time of Algorithm 1, a major contributing factor is the computation of the $(p-q)\sqrt{n}/\sqrt{p(1-q)}$ dimensional SVD projection, which can be significantly more efficient than the previous state-of-the-art algorithm by Ailon, Chen, and Xu (JLMR 2015). The latter involves solving a convex programming problem with $n^2$ constraints, rendering it computationally prohibitive.
>
> For instance, in experiment 3A of the JLMR 2015 paper, the authors mentioned that for an instance with $n=3500$ and $k=4$, they required $182$ seconds to recover the clusters. In contrast, our method finishes in approximately $2.5$ seconds. Thus, this is another empirical improvement over the state-of-the-art algorithms!
>
>
> ### Spectral Clustering:
> Spectral clustering is indeed a powerful tool. However, in this specific problem, some instances may be hard. For example, we ran a version of spectral clustering for the parameters of  Experiment 5 of Table 2 of our paper, both with $K=2$ (which is the number of large clusters) as well as $K=1000$ (which is the number of total clusters). The algorithm failed to recover a ground truth cluster in either case.

---

> > ### Comment · Reviewer_MiyQ · 2023-08-10
> > **Response**
> >
> > Thank you for your response and clarification, I keep my positive evaluation of the paper.

---

> > > ### Author Response · Authors · 2023-08-14
> > >
> > > Thanks again for your detailed review and considering our responses!

---

### Official Review · Reviewer_tK15 · 2023-07-06

**Soundness:** 3 good
**Presentation:** 4 excellent
**Contribution:** 2 fair
**Rating:** 5
**Confidence:** 1

**Summary:**

The authors consider the problem of perfect recovery in a stochastic block model where the average degree is large and where the groups are not balanced. They provide an algorithm based on singular value decomposition to recover recursively the largest clusters. They provide a few numerical experiments illustrating their claims. The authors apply their results to the problem of clustering with a faulty oracle.

**Strengths:**

I have little knowledge as to this problem of perfect recovery in a dense SBM and I am not able to assess the correctness of the claims and their relevance.


**Weaknesses:**

The same.

**Questions:**

Maybe the authors could precise the complexity of the algorithm 1. In experiment 6 it seems the authors are able to run this algorithm for n substantially larger than the other experiments. The authors could go to higher n and test how tight are their bounds; in particular taking p and q smaller.

A small section to conclude the article and for future work would be appreciable.

Some references are ill-formatted. Eg ref. 27 "svd" –> "SVD".
Inconsistency: plural set vs plural-set.

**Limitations:**

The same.

---

> ### Author Rebuttal · Authors · 2023-08-09
>
> We thank you for your efforts! We will definitely add a concluding section highlighting some future works that we think will be of interest.

---

> > ### Comment · Reviewer_tK15 · 2023-08-14
> >
> > I thank the authors for their answers.

---

### Official Review · Reviewer_Ajy4 · 2023-07-07

**Soundness:** 3 good
**Presentation:** 3 good
**Contribution:** 2 fair
**Rating:** 5
**Confidence:** 4

**Summary:**

The paper deals with the problem of community detection for unbalanced community sizes. Specifically, the paper concentrates on the situation where both large (O(\sqrt{n})) and small communities exist in the network. The paper proposes a stepwise method of recovering the large clusters in the presence of small clusters for planted clique SBM and faulty oracle models.

**Strengths:**

The main strengths of the paper are as follows -

(1) The paper addresses a gap in the literature on the simultaneous recovery of large and small communities in networks.

(2) The paper deals with the problem of community recovery of large communities in the presence of small communities. The paper provides a stepwise method of recovering large communities in planted clique SBM and faulty oracle models.

(3) The paper provides theoretical results supporting the recovery of large communities by overcoming the "small cluster barrier" of the size of the remaining small clusters.

(4) The paper is well-written.

**Weaknesses:**

The main weaknesses of the paper are as follows -

(1) The paper misses some relevant literature. Such as - Li, Tianxi, et.al. "Hierarchical community detection by recursive partitioning." Journal of the American Statistical Association 117, no. 538 (2022): 951-968. It describes an algorithm that is very similar to the algorithm proposed in this work.

(2) Algorithms 2 and 3 assumes the knowledge of p and q, which are very strong assumptions. It is not immediately clear how the algorithm can be extended for general SBM.

(3) The stopping criterion of the proposed algorithm is not clear.


**Questions:**

(1) Does the proposed algorithm assume the knowledge of p and q?

(2) Does the proposed algorithm assume the knowledge of the number of communities, or is there a stopping criteria of the proposed algorithm for recovery of the number of large communities?

---

> ### Author Rebuttal · Authors · 2023-08-09
>
> We thank you for your detailed review. Please find our comments on the weakness and the answers to your question below.
>
>
> ### Knowledge of $p,q,k$:
> Our algorithm necessitates knowledge of the parameters $p$ and $q$ but not the number $k$ of communities. However, it is worth noting that even the previously state-of-the-art algorithms by Ailon, Chen, and Xu (ICML 2013 and JMLR 2015) **also** rely on knowing $p, q$, as well as $k$ (or in some special cases, an upper bound on $k$). This observation highlights the inherent difficulty of theoretically addressing the ``small cluster barrier.'' An open question arises: Can our algorithm be made parameter-free? Exploring this possibility remains an interesting direction for future research.
>
>
> ### Stopping Criterion of the algorithm:
>
> 1) The *EstimatingSize* subroutine (Algorithm 3) calculates $\bar{s}$, which is approximately
> $
> \max ( 100\sqrt{p(1-q)}\sqrt{n} \log n/(p-q), 0.5\cdot s_{\max} ).
> $
>
> Within the *RecursiveCluster* algorithm, in each iteration, we can check if the $\bar{s}$ value is sufficiently large and use that as the stopping criterion to determine if the algorithm should halt its iterative process.
>
>
> 2) To address the confusion of the unclear stopping criteria, we will add an explanatory line following line 2 in Algorithm 1, stating that if *EstimatingSize* does not return a valid $\bar{s}$ (i.e. when Exit(0) occurs in Algorithm 3), then Algorithm 1 will simply return an empty set.
>
>
> ### Relevant literature:
> We appreciate the reference to the paper (where the analysis is on the binary tree SBM model) you mentioned, and upon careful examination, we have noticed some fundamental differences between the referred algorithm and ours, despite being related. (We will also add the comparison to this relevant literature in the future version.)
>
> i) The algorithm in the mentioned paper utilizes the well-known eigenvector+K-Means approach, whereas our approach involves a distinct SVD+ ``plural-set-purification'' strategy.
>
> ii) Notably, the binary tree SBM problem comprises small clusters as "sub-clusters" within larger clusters, whereas in the SBM model (as studied in our paper), very small clusters can be considered as potential noise as the cluster sizes can be arbitrarily unbalanced.
>
> iii) Furthermore, it appears that their algorithm requires knowledge of the number of sub-clusters at each level (they can estimate whether there are any more levels), which is inherently different from the problem we are addressing, and we do not require the knowledge of cluster sizes and number of clusters.
>
>
> ### Extension to general SBM:
> To extend our findings to the general SBM, we propose adapting our algorithm by setting $p$ to represent the minimum intra-cluster probability and $q$ to denote the maximum inter-cluster probability. In a future extended version, we plan to explore this in depth. To achieve this, we will devise a more refined and nuanced method for estimating the largest cluster size. Then by leveraging the aforementioned approach, we aim to identify the largest cluster within general SBM efficiently.

---

> > ### Comment · Reviewer_Ajy4 · 2023-08-18
> >
> > Thanks for the detailed rebuttal comments.
> >
> > I am also increasing the score in response to the explanations.

---

> > > ### Author Response · Authors · 2023-08-18
> > >
> > > Thank you for your response and consideration.

---

### Decision · Program_Chairs · 2023-09-21

**Decision:**

Accept (poster)

**Comment:**

The paper studies the exact recovery problem in the SBM when the input community are unbalanced. This setting introduces some interesting technical challenges and it received a bit less attention in the literature(despite being probably the most realistic scenario).

In this setting the paper is a nice improvement on the state-of-the-art. Furthermore, in contrast with previous work the authors do not need to know the number of clusters as an input parameters.

Overall, the paper will be a nice addition to the conference program.

The reviewers have some interesting suggestions that the authors should address in the final version. For example:
- adding experiments on real world data
- adding runtime experimental benchmark
- mention and explain idea of how the 2^13 factor could be improved